



# Grazing-related nitrous oxide emissions: from patch scale to field scale

**Karl Voglmeier[1,2], Johan Six[2], Markus Jocher[1], and Christof Ammann[1]**

[1]Climate and Agriculture Group, Agroscope, Zürich, 8046, Switzerland
[2]Department of Environmental Systems Science, ETH Zurich, Zürich, 8092, Switzerland

**Correspondence:** Christof Ammann (christof.ammann@agroscope.admin.ch)

**Abstract.** Grazed pastures are strong sources of the greenhouse gas nitrous oxide ($N_2O$). The quantification of $N_2O$ emissions is challenging due to the strong spatial and temporal variabilities of the emission sources and so $N_2O$ emission estimates are very uncertain. This study presents $N_2O$ emission measurements from two grazing systems in western Switzerland over the grazing season of 2016. The 12 dairy cows of each herd were kept in an intensive rotational grazing management. The diet for the two herds of cows consisted of different protein-to-energy ratios (system G: grass only diet; system M: grass with additional maize silage) resulting in different nitrogen (N) excretion rates. The N in the excretion was estimated by calculating the animal nitrogen budget taking into account the measurements of feed intake, milk yield, and body weight of the cow herds. Directly after the rotational grazing phases, background and urine patches were identified based on soil electric conductivity measurements while fresh dung patches were identified visually. The magnitude and temporal pattern of these different emission sources were measured with a fast-box (FB) chamber and the field-scale fluxes were quantified using two eddy covariance (EC) systems. The FB measurements were finally upscaled to the field level and compared to the EC measurements for quality control by using EC footprint estimates of a backward Lagrangian stochastic dispersion model. The comparison between the two grazing systems was performed during emission periods that were not influenced by fertilizer applications. This allowed the calculation of the excreta-related $N_2O$ emissions per cow and grazing hour and resulted in considerably higher emissions for system G compared to system M. Relating the found emissions to the excreta N resulted in excreta-related emission factors (EFs) of $0.74 \pm 0.26$ % for system M and $0.83 \pm 0.29$ % for system G. These EF values were thus significantly smaller compared to the default EF of 2 % provided by the IPCC guidelines for cattle excreta deposited on pasture. The measurements showed that urine patch emission dominated the field-scale fluxes (57 %), followed by significant background emissions (38 %), and only a small contribution of dung patch emission (5 %). The resulting source-specific EFs exhibited a clear difference between urine ($1.12 \pm 0.43$ %) and dung ($0.16 \pm 0.06$ %), supporting a disaggregation of the grazing-related EFs by excreta type in emission inventories. The study also highlights the advantage of a N-optimized diet, which resulted in reduced $N_2O$ emissions from animal excreta.

# 1 Introduction

Nitrous oxide ($N_2O$) is a strong greenhouse gas (GHG) with a 265 times stronger warming potential compared to $CO_2$ on a mass basis (IPCC, 2014). Typically an inert gas in the troposphere, $N_2O$ has a strong potential to destroy the ozone layer in the stratosphere (Portmann et al., 2012). The largest share of $N_2O$ emissions are attributed to nitrogen (N) fertilization in the agricultural sector, but livestock grazing, especially by cows, can also lead to significant direct and indirect $N_2O$ emissions due to excreta from the animals (Luo et al., 2017; Reay et al., 2012). The nitrogen deposited by animal excreta often exceeds the N applied by fertilizer (Aarons et al., 2017). The available reactive N is used by microbial nitrification and denitrification processes where significant amounts of $N_2O$ can be produced (Selbie et al., 2015). A nonlinear response of $N_2O$ emissions to N loading has

been shown previously (Cardenas et al., 2010), and urine patches of cattle have exceptionally high N loading rates (up to $2000 \, \mathrm{kg \, N \, ha^{-1}}$), making them especially prone to high $N_2O$ losses (Selbie et al., 2015).

For inventories and live cycle assessments, the magnitude of the $N_2O$ emissions is usually calculated by applying emission factors (EFs) related to the magnitude of N inputs to the agricultural fields (EF = emitted $N_2O$-N/N input). According to the guidelines of the Intergovernmental Panel on Climate Change (IPCC, 2006) for national emission reporting, a separation is made between (i) emissions related to excreta N deposited by the grazing animals and (ii) emissions related to fertilizer applications and other N inputs. While for fertilizer-induced $N_2O$ emissions, a default value of 1 % is proposed by IPCC (2006), the default EF related to excreta of grazing cattle (denoted as $EF_{3PRP,CPP}$) is 2 %. Most countries including Switzerland presently use these default values. However, the default $EF_{3PRP,CPP}$ value often overestimates observed pasture emissions (Bell et al., 2015; Chadwick et al., 2018) and does not take into account country-specific conditions (climate, soil, management). Therefore, some countries have developed a country-specific EF (e.g., New Zealand; Saggar et al., 2015) which is still lacking for Switzerland. Additionally, it has been shown that separate EFs for urine and dung might be beneficial in describing the emissions and understanding the contributions of the different emission sources on a pasture (Bell et al., 2015). A better understanding of the individual contributions would also be very helpful to reduce the emissions, as dietary changes, for example, typically affect the excreted urine N, which is mainly responsible for the high $N_2O$ emission associated with excreta (Dijkstra et al., 2013). However, the range and thus the uncertainty of specific urine EFs is rather large (0 %–14 %, $n = 40$) as shown by Selbie et al. (2015) based on a survey of literature reports. Many of those studies measured the emissions on artificially applied urine or under laboratory conditions, making these results questionable with regard to the applicability within greenhouse gas inventories.

The efficient use of fed N is essential to reduce the emissions associated with animal excreta. Studies have shown that an optimized feeding strategy can lead to less N excreted by the animals (e.g., Arriaga et al., 2010; Dijkstra et al., 2013; Yan et al., 2006). For this purpose, forage with a low N content (e.g., maize) can be used as a supplement to N-rich grass and this subsequently leads to less N in the excreta, mainly in the form of less urine N. A lower amount of N input to the pasture is supposed to produce less $N_2O$ emissions, but corresponding emission experiments under real grazing conditions for a full season, to our knowledge, have not been reported hitherto.

Historically, most studies used static chambers to quantify $N_2O$ emissions (Flechard et al., 2007). Chamber measurements are ideal to quantify emissions on a small spatial scale and to attribute the measured fluxes to certain emission drivers, but for excreta emissions these measurements were often performed on manually applied urine and dung patches (Bell et al., 2015; Cai and Akiyama, 2016). Additionally, due to the strong heterogeneity of the emissions from a pasture (Cowan et al., 2015; Flechard et al., 2007) chamber techniques are not ideal to compute field-scale emissions for grazing systems. The eddy covariance (EC) method overcomes this problem by integrating over multiple emission sources over a larger spatial domain. The EC technique was already applied successfully to quantify $N_2O$ emissions from pastures and grasslands (Jones et al., 2011). Some studies also tried to compare different systems (e.g., intensive – extensive, different crops, land/lake) with one EC tower (e.g., Biermann et al., 2014; Fuchs et al., 2018) by partitioning the fluxes based on wind direction and system geometry, but typically one tower for each system is preferable. In order to understand and quantify the emissions of a pasture, the combined approach of EC measurements and chambers is regarded as the best solution (Cowan et al., 2015). The EC systems can be used to quantify the field-scale emissions while the chamber approach can be used to estimate the contributions from single emission sources (urine patches, dung patches, and other "background" areas).

In our experiment, we measured $N_2O$ emissions from two neighboring pastures simultaneously with the EC method over a full grazing season. The two pastures differed in the energy-to-protein balance of the cows' diet. The small-scale fluxes were quantified with a fast-box chamber and upscaled to match the EC flux footprints for comparison. Further on, we computed the contribution of the different emission sources to the overall pasture emissions. The results were compared to default values provided by IPCC and other literature values. The main goal of the study was to quantify the excreta-related emission and the corresponding EF for real grazing systems and to analyze the specific contributions of dung and urine patches.

## 2    Material and methods

### 2.1    Experimental site

The experiment was conducted at the research farm Agroscope Posieux in the Prealps of Switzerland in the canton of Fribourg ($46°46'04''$ N, $7°06'28''$ E) during the grazing season of 2016 and has already been described in detail by Voglmeier et al. (2018). The farm is located at an elevation of 642 m with an annual average temperature of $8.7 \, °C$ and a mean annual precipitation sum of 1075 mm (MeteoSwiss, 2018). The soil consisted mainly of a stagnic Anthrosol with a loamy texture (see Table 1). Soil profile samples for analysis of texture and other soil characteristics were taken at four locations on the pasture in 2013 and 2016. The vegetation consisted of a grass–clover mixture typical for Swiss pastures ($78 \pm 12$ % grasses and $15 \pm 10$ % legumes; main species: *Lolium perenne* and *Trifolium repens*, 10 sampling

**Table 1.** Near-surface soil parameters (5–10 cm depth) averaged over four locations on the pasture. The measurements are given as mean $\pm 1$ standard deviation.

| Parameter | Value |
|---|---|
| Pore volume (%) | $57 \pm 4$ |
| Bulk density (g cm$^{-3}$) | $1.09 \pm 0.11$ |
| pH (–) | $6.0 \pm 0.3$ |
| Sand (%) | $42.6 \pm 2.5$ |
| Clay (%) | $18.7 \pm 1.7$ |
| Silt (%) | $33.0 \pm 1.3$ |
| Soil organic matter (%) | $5.7 \pm 0.3$ |
| Total N (%)* | $0.38 \pm 0.03$ |
| Total C (%)* | $3.76 \pm 0.20$ |

\* Measured at a depth of 0–10 cm.

**Table 2.** Measured averages $\pm$ standard deviation of observed cow properties (ECM: energy corrected milk) and feed protein contents used by the dairy cow nitrogen budget approach for both pasture systems during the grazing season 2016.

| Input parameter (units) | System M | System G |
|---|---|---|
| Number of cows | 12 | 12 |
| Milk yield, ECM (kg cow$^{-1}$ day$^{-1}$) | $25.1 \pm 2.9$ | $24.2 \pm 3.7$ |
| Animal weight (kg) | $633 \pm 14$ | $633 \pm 10$ |
| Grass crude protein (g kg-DM$^{-1}$) | $195 \pm 23$ | $196 \pm 23$ |
| Maize crude protein (g kg-DM$^{-1}$) | $84 \pm 8$ | n/a |

n/a – not applicable

times between May and September). After the last renovation treatment in 2007 the field had been used as an intensive pasture for cattle grazing with occasional grass cuts for maintaining a homogenous sward. In addition to the N input through excreta from the grazing animals, N had been applied through fertilizer at a rate of about 120 kg N ha$^{-1}$ per year between 2007 and 2015.

## 2.2 Experimental design

The experiment took place at a 5.5 ha pasture, which was divided into two separate systems differing in feeding strategy of the 12 cows per system (Fig. 1a). The northern system (system M) represented a N-optimized feeding option where the diet of the cows consisted of grass with additional maize silage (roughly 20 % of the dry matter intake, DMI, fed in barn during milking periods) resulting in a demand-optimized protein content in the diet (Arriaga et al., 2010; Yan et al., 2006). This was supposed to reduce the excreta N input to the pasture. The southern system (system G) represented a full grazing regime with no additional forage, which resulted in a considerable protein surplus (see Table 2). Both systems were managed as a rotational grazing system with 11 paddocks (Fig. 1a) resulting in a typical rotation period of about 20 days. The size of the paddocks was adjusted for the different feeding strategies and resulted in typical sizes of 1700 m$^2$ for system M and 2200 m$^2$ for system G. The rotation of both systems was managed synchronously with a new rotation starting on the westerly paddocks ($X$.11 to $X$.16 with $X$ indicating both systems) followed by the easterly ones ($X$.21 to $X$.25).

Grazing on the paddocks started with intermittent grazing phases in March and ended in early November with the main grazing season being between the end of April and early October. During this time period eight full rotations took place. The cows typically spent 18 to 20 h per day on the pasture and were brought to the barn twice a day (around 05:00 and 17:00 LT) for milking. However, in July and August the cows spent a longer time in the barn during daytime (up to 6 h; see Fig. 2c) mainly due to high air temperatures and to a minor degree to additional experiments of other research groups. Heavy rain events in June led to very wet soil conditions, which prevented grazing between 8 June and 4 July and necessitated a grass cut on 22 and 27 June (Fig. 2c).

## 2.3 N input to the pasture

During the grazing season, N input to the pasture mainly occurred in the form of excreta of the grazing animals and to a lesser extent as mineral fertilizer (Fig. 2d). The mineral fertilizer was ammonium nitrate (28 kg ha$^{-1}$) applied at the end of June and urea (42 kg ha$^{-1}$) with a split application between the middle of August (western paddocks $X$.11–$X$.16) and early September (eastern paddocks $X$.21–$X$.25) due to concurrent grazing. In the present study we focus on the N input by grazing excreta and their effect on N$_2$O emissions. The comparison between the field-scale EC method and the small-scale chamber measurements also required estimates of the number of dung and urine patches on the pasture. These numbers were calculated as described in Sect. 2.7 based on the excreted N amounts. N excretion cannot easily be measured in the field, but it can be calculated based on the energy demand of the cows and measured N in feeds and products (e.g., milk, body weight gain). We followed the approach described by Felber et al. (2016) to calculate the energy and N flows of the dairy cows in the experiment and to calculate daily values of excreted N per cow. Input parameters to the budget calculation were daily measurements of milk yield, milk N content, and body weight gain as well as seasonal measurements of protein content of the grass (eight times between end of April and end of September) and of the maize silage (three times between beginning of May and beginning of September). The breakdown of the excreted N in urine and dung was based on work by Bracher et al. (2011). For further details see Voglmeier et al. (2018), where the corresponding uncertainty of the total N and urine/dung N was estimated to be 15 % ($2\sigma$) for the same experiment. Seasonal statistics of the input variables are given in Table 2.

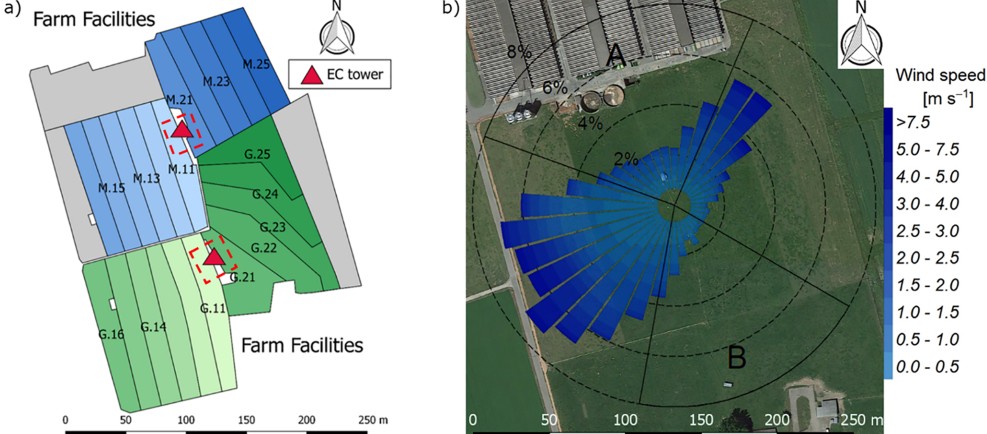

**Figure 1. (a)** Measurement site with the pastures for the two herds (blue: grass diet with additional maize silage; green: full grazing regime; grey: optional pasture areas) and the division into the paddocks (M.11–M.25, G.11–G.25). Additionally the location of the two EC towers (triangles) and the area of the chamber measurements (red dashed rectangles) are shown. **(b)** Wind distribution for the northern sonic anemometer with the corresponding sector contributions (black dotted circles) for the period May–October 2016. The areas A and B indicate wind sectors from which advection from nearby farm building can occur. The wind distribution was overlaid on a Google Earth image of the experimental area (map data: Google, DigitalGlobe).

## 2.4 Small-scale flux measurements

### 2.4.1 Excreta detection

The localization of fresh dung and urine patches was essential in this study to measure $N_2O$ emissions attributable to specific excreta sources. Intensive observation areas of $10\,\mathrm{m} \times 10\,\mathrm{m}$ or $15\,\mathrm{m} \times 15\,\mathrm{m}$ close to both EC towers in the paddocks $X.11$ and $X.21$, respectively (see Fig. 1a) were selected. Within these areas fresh dung and urine patches were mapped typically 1–3 days after grazing of the respective paddock. Dung pats were mapped visually and labeled for subsequent chamber measurements. For urine patches a direct visual identification was not possible. Bates et al. (2015) demonstrated the ability of surface-soil electrical conductivity measurements to detect urine patches. Using this approach we mounted a soil probe (GS3, Meter Group, US; for soil moisture, temperature, and electrical conductivity measurements) on a handheld stick and mapped the intensive observation area on a 25 cm grid (Fig. 3). Based on pre-experimental tests, areas with conductivity values below a threshold of $0.15\,\mathrm{mS\,cm^{-1}}$ (dark blue areas in Fig. 3a) were considered background without recent influence of excreta. Spots with a conductivity above the threshold were marked as possible urine patches for the chamber measurements. Time series of electrical conductivity measurements (Fig. 3b) on manually applied urine patches in 2017 illustrate the long-term effect and demonstrate the possibility to distinguish between background areas and urine patches more than 10 days after the application of urine.

### 2.4.2 Fast-box measurements

Small-scale emissions from urine and dung patches as well as background pasture areas were measured with a fast-box (FB) chamber (Hensen et al., 2006). The measurements took place on the paddocks $X.11$ and $X.21$ (Fig. 1a) between the beginning of July and middle of October and were therefore taken mainly during dry soil conditions (Fig. 2a, periods with $VWC < 0.4$). Measurements usually started after the excretion detection (Sect. 2.4.1) and about 1–2 days after the end of grazing (EOG). The age of the excreta patches is important for the interpretation of the measured fluxes. However, the exact determination of the excreta age was not possible. Thus, the time since EOG was used as excreta age for each FB measurement. The potential age variability of a single excreta patch resulted from the sojourn time of the cows on the paddock, which typically was in the range of 1–1.5 days.

The manually operated opaque $0.8\,\mathrm{m} \times 0.8\,\mathrm{m} \times 0.5\,\mathrm{m}$ box was connected to a fast response quantum cascade laser analyzer (QCL, Aerodyne Research Inc.) that was also used for the EC system on the respective field (see below Sect. 2.5.1). The sample air was drawn continuously from the FB headspace through a 40 m 1/4″ polyamide (PA) tube to the analyzer, allowing measurements within a radius of about 35 m on the paddocks $X.11$ and $X.21$ (see Fig. 1). The sample flow rate $Q$ was typically around $8\,\mathrm{L\,min^{-1}}$. The box was modified by using a defined vent to ambient air through a tube of 4 cm in diameter and 1 m in length. The inlet of the vent tube was packed with a foam material over a length of 10 cm to avoid uncontrolled air exchange due to wind-induced pressure fluctuations. The chamber was also equipped with a GMP343 $CO_2$ probe (Vaisala, FI) to mea-

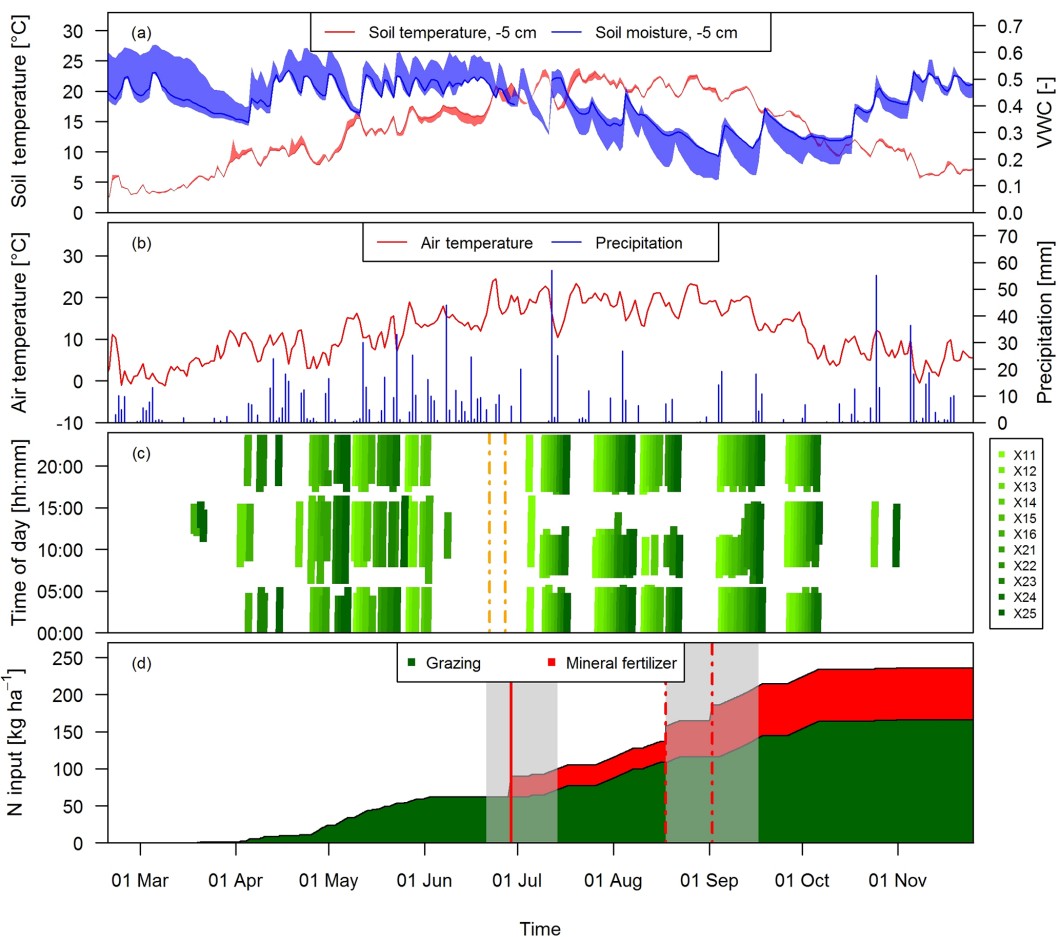

**Figure 2.** Time series of **(a)** daily averaged soil temperature and moisture at a depth of 5 cm measured at system M (solid lines) and spread of the four measurement locations, **(b)** daily air temperature at 2 m above ground and precipitation at the measurement site, and **(c)** grazing duration on the single paddocks of the pasture ($X$: both pasture systems M and G) for the study year 2016. The dashed vertical orange lines indicate the harvest event (split between $X.11$–$X.16$ and $X.21$–$X.25$) **(d)** N input to system M during the main grazing season in 2016. Fertilizer was applied two times (vertical red lines). The second application (dashed lines) in August was split in two parts due to concurrent rotational grazing. The grey shaded areas indicate time periods influenced by fertilization or harvest events as explained in Sect. 2.5.2.

sure the soil respiration, which was used for quality control purposes (Sect. 2.4.3). The increase in N$_2$O concentration after placing the chamber on the soil with a flux $F_{\text{Cham}}$ was recorded every 3 s for a time period of about 90 s (taking into account the time delay due to tube sampling). The inflow of the background concentration $C_{\text{bg}}$ into the chamber volume $V$ (with area $A$) through the vent led to lower measured concentration values $C$. This can be described by the following differential equation for the chamber headspace concentration $C(t)$:

$$V \frac{\delta C}{\delta t} = A \cdot F_{\text{Cham}} - Q \left( C - C_{\text{bg}} \right). \tag{1a}$$

This is a combination of the two equations for static chambers (right-hand term $= 0$) and for the dynamic chamber (left-hand term $= 0$). Solving of the equation yields the explicit time function

$$C(t) = \frac{A \cdot F_{\text{Cham}}}{Q} \left( 1 - e^{-\frac{Q}{V} \cdot t} \right) + C_{\text{bg}}. \tag{1b}$$

For small values of the exponent $Q/V \cdot t$ (slow chamber volume exchange of about 40 min and short measurement time) as characteristic for the present fast-box measurements, the entire bracket term can be linearized with a series expansion to $(Q/V \cdot t)$. Inserting the resulting function for $C(t)$ into Eq. (1a) yields

$$V \frac{\delta C}{\delta t} = A \cdot F_{\text{Cham}} \left( 1 + \frac{Q}{V} t \right). \tag{1c}$$

With the FB dimensions and sampling flow rate as given above and a maximum accumulation time $t \leq 2$ min, the deviation from the ideal linear increase in a fully closed static chamber was $\leq 5\%$. The flux was finally calculated by using

the HMR package (Pedersen et al., 2010), which uses linear and nonlinear regression to fit the measured concentration values. The uncertainty of an individual box measurement is estimated to be around 20 % (Hensen et al., 2006).

In order to relate the measured fluxes to environmental driving parameters the following sensors were placed inside on the chamber: a thermocouple (type K) for air temperature measurement within the chamber, a GS3 probe (see Sect. 2.4.1) for soil moisture, soil temperature, and soil conductivity measurements (ca. 0–5 cm depth), and an ML3 ThetaProbe (Delta-T Devices Ltd., UK) for soil moisture and soil temperature observations (ca. 0–10 cm depth). All measured data values were stored on a data logger mounted on top of the box and transferred to a computer in the nearby shelter or trailer. A customized LabVIEW (National Instruments, US) program allowed for online inspection of all measured data values including the gas concentrations.

### 2.4.3   Quality control and system comparison

FB fluxes were selected for post-processing after fulfilling certain quality criteria. In a first step, the $R^2$ value of any flux calculation had to exceed 0.9 (e.g., for $N_2O$ flux the $R^2$ value of $N_2O$, $CH_4$, or $CO_2$ had to exceed 0.9). For urine patches, the soil conductivity had to exceed $0.25\,\mathrm{mS\,cm^{-1}}$ at the beginning of the measurements (see also Fig. 3b) in order to exclude possible old urine patches (of previous management rotations). Presumable old patches were therefore rejected for further processing. Background fluxes were removed from further processing if the flux value exceeded $40\,\mathrm{\mu g\,m^{-2}\,h^{-1}}$ ($= 4\times$ median value) to ensure that undetected urine patches at the chamber surroundings did not influence the flux measurements. Finally, 360 and 293 flux measurements met the criteria on systems M and G, respectively. These measurements were composed of 238 background fluxes, 242 urine patch fluxes, and 173 dung fluxes.

For a direct comparison of the FB measurements on the two pasture systems, the fluxes obtained on the same day were ordered based on their magnitude for each system and source class. Due to the synchronous grazing regime, the fluxes represented the same excreta age (e.g., on day 3 after EOG). However, synchronous FB measurements on both systems were not always performed. Resulting numbers of data pairs are 46, 54, and 40 for background, urine, and dung fluxes, respectively.

### 2.5   Field-scale flux measurements

#### 2.5.1   Eddy covariance system

For field-scale flux measurements EC towers were installed in the middle of the two pasture fields to account for the predominant wind directions northeast and southwest (Fig. 1) and were fenced with a radius of 2–3 m to avoid unwanted animal contact. The measurement height was 2 m, which enabled a good footprint coverage (Fig. 4, Sect. 2.5.4) of both fields and allowed the measurement of field-scale fluxes of both systems.

The two EC systems were identically equipped with an ultrasonic anemometer–thermometer (further on named sonic, HS-50, Gill Instruments Ltd., UK) to quantify the turbulent mixing by measuring the three-dimensional wind velocity ($u, v, w$) and air temperature. Dry air mixing ratios of $N_2O$ were measured with closed-path quantum cascade laser spectrometers (QCL, QC-TILDAS, Aerodyne Research Inc.) that analyzed air samples drawn through a 25 m PA tube (inner diameter 6 mm) by a vacuum pump (Bluffton Motor Works, flow rate ca. $13\,\mathrm{L\,min^{-1}}$). One filter at the inlet (AcroPak, Pall Corporation, 0.2 μm) and one before the instrument (Midisart 2000, Sartorius Stedim Biotech GmbH, 0.2 μm) were used for each system to filter out particles. The distance of the inlets of the QCL from the center of the sonic head were around 20 cm and the QCL instruments were placed in a temperature-controlled environment (trailer at system M, shelter at system G) about 20 m north (system M) or south (system G) of the EC towers.

The sample frequency of the EC system was 10 Hz. A customized LabVIEW (National Instruments, US) program was used to combine the data strings of the individual instruments and store them as binary raw data for offline analyses. Additionally the program visualized the measurements and fluxes of the $N_2O$ concentrations and fluxes, calculated with an online flux calculation. The program also allowed us to check the EC system by remote access.

#### 2.5.2   Flux calculation

A customized program written in the statistical software R (R Core Team, 2016) was used to calculate EC fluxes for 30 min intervals (similar to Felber, 2015, and Felber et al., 2015). The program is based on Ammann et al. (2006, 2007). In a first step, 10 Hz data outside a plausible physical range were identified and replaced by a running mean filter with a window size of 500 data points. In a next step, wind vector components were rotated into the mean wind direction using the double coordinate rotation technique (Kaimal and Finnigan, 1994), and concentration values were subject to linear detrending within an averaging interval of 5 min.

The EC flux is defined as the covariance of the vertical wind speed and the trace gas mixing ratio. Due to the long inlet tube the time series of the trace gas signals are delayed in relation to the wind measurements by a quasi-constant lag time of about 6 s for system M and 7 s for system G. Thus, the trace gas signals have to be shifted to obtain the correct covariance flux (Langford et al., 2015). In a pre-evaluation, the "default lag" was determined as the most frequent position of the maximum absolute value of the cross-covariance function over periods of weeks to months (depending on instrument maintenance). Then it was checked for each half-hour period whether the individual "dynamic" lag was within

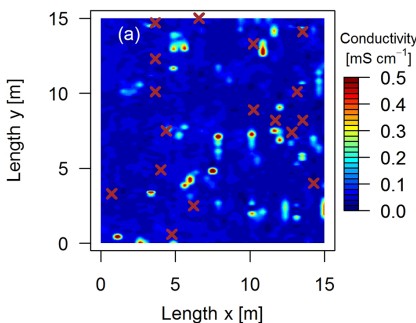
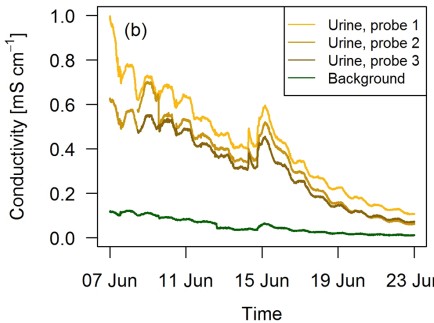

**Figure 3. (a)** Measured conductivity within a quadratic 15 m × 15 m intensive observation area on the 3 October 2016 in system G. High values ($> 0.15$ mS cm$^{-1}$) indicate urine patch locations and brown crosses indicate observed dung pats. **(b)** Conductivity measured continuously during a field experiment in 2017 with four GS3 sensors.

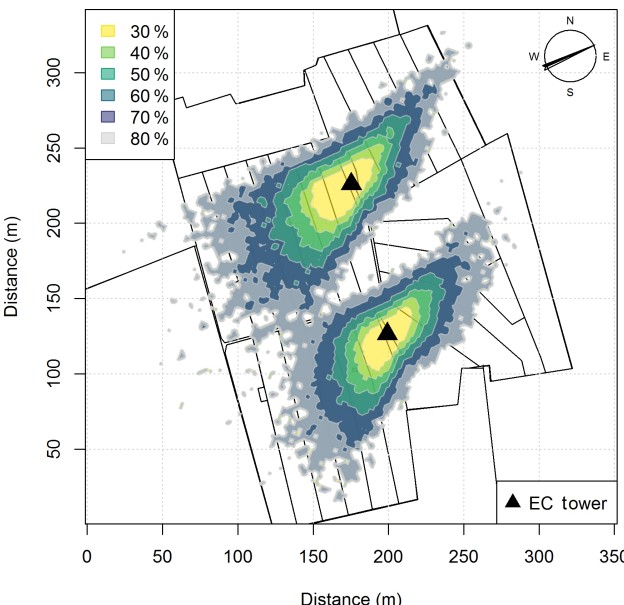

**Figure 4.** Footprint climatology for both EC towers averaged for the time period between 15 March and 15 November 2016. The legend values indicate the percentage of the total footprint weight.

were quantified using the "ogive" method where the damping factor was calculated by fitting the normalized cumulative co-spectrum of $N_2O$ to the one of the sensible heat at a frequency of 0.065 Hz. In a post-processing step, these half-hourly damping factors were filtered for favorable conditions e.g., low noise level of the ogive and the flux. The selected values were used to compute a damping function dependent on wind speed and stability which was finally used to estimate the damping factor. Depending mainly on the wind speed, a damping effect of 10 %–30 % was found and corrected for.

EC fluxes were measured continuously over the grazing season. Since the present study is focused on $N_2O$ emissions from grazing, time periods with strong influence of $N_2O$ emissions from fertilization and harvest events (see Fig. 2c–d) were excluded for computation of cumulative emissions and for comparisons between field-scale and small-scale measurements. These exclusion periods were limited to the 15 days following fertilization or harvest and led to a rejection of 47 days during the grazing season. The criterion is based on observed EC fluxes (Sect. 3.1) and is in accordance with Jones et al. (2011). The time period used for calculation of the cumulative grazing emissions is further on defined as grazing-only periods (GOPs) and accumulated to 198 days.

### 2.5.3 Quality control and gap filling

EC flux measurements are subject to different sources of measurement problems and quality issues, which often result in data loss or data rejection. These sources can be instrument specific like power failures or malfunctioning, environmental driven like measurements under nonideal conditions (e.g., low turbulence), or a combination of both (Papale, 2012). Power outage, instrument maintenance (only on system M), and delayed installation (only on system G) led to data losses during the GOP of 12 % and 17 % for systems M and G, respectively. Data rejection due to low friction velocity ($u_* < 0.07$ m s$^{-1}$) and large vertical tilt angle ($-2$ to $6°$) of the wind vector led to a further data loss of about 35 %.

a time window of 0.61 s around the default lag. If this was the case, the dynamic lag was used, otherwise the default lag was used. In order to minimize the effect of nonstationarities in the time series, the 30 min flux was finally calculated as the average over six 5 min subinterval flux values. This caused a minor low-frequency spectral loss (1 %–5 %) that was quantified (and corrected for) using Kaimal cospectra and the theoretical transfer function for block averaging.

The fluxes measured by EC systems are also subject to different high-frequency losses due to sensor separation and in case of $N_2O$ air transport through the inlet tubes (Foken et al., 2012). These damping effects can lead to a significant underestimation of the flux and must be corrected. Based on Ammann et al. (2006) the half-hourly high-frequency losses

Because nonstationarity of the flux was already reduced by the short averaging/detrending interval of 5 min, a quality selection based on nonstationarity (Foken et al., 2012) had little effect and was therefore not used here. Additional rejection of wind sectors influenced by the farm facilities, trailer, or shelter and to avoid cross-influences from the other pasture system (wind dir = 280–25° and wind dir = 97–195°) contributed to an overall data loss of 64 % and 69 % for systems M and G. The resulting occurrence of data gaps showed a diurnal pattern with stronger data loss during the night, which was driven by the wind pattern with typically stronger wind speeds during daytime and calm nights.

The gaps in the flux time series needed to be filled in order to compute cumulative sums over a certain period of time. However, no well-established reference method for the gap filling of $N_2O$ fluxes exists to date. We followed the evaluation of Mishurov and Kiely (2011) and used a lookup table method (LUT) with three parameters: one for the preceding cumulative rainfall of the last 12 h with three classes (no rainfall, 0–2 mm, > 2 mm), one for the percentiles of the soil temperature at 5 cm depth during the GOP with four classes (0–25th percentile, > 25th percentile–median, > median–75th percentile, > 75th percentile), and one for the footprint-weighted (Sect. 2.5.4) averaged cow density (cows ha$^{-1}$) on the single paddocks over the preceding 5 days (0, 0–2, > 2 cows ha$^{-1}$). To check the sensitivity towards different gap-filling methods, three other techniques were compared to the LUT approach: (I) running mean with a variable filter window size and at least 12 values, (II) monthly mean diurnal variation (MDV; see Zhao and Huang, 2015) with a running half-hourly window size of five in order to have more values during nighttime, and (III) seasonal MDV based on half-hourly values averaged over the whole grazing season. Due to the delayed installation of the EC tower on the southern field, all values prior to 14 April on system G resulted from the gap-filling routine. The uncertainty of gap filling for seasonal cumulative fluxes was estimated from the standard deviations of monthly cumulative fluxes retrieved with the different gap-filling methods during GOP, which resulted in an uncertainty of 14 % and 18 % for systems M and G, respectively (1$\sigma$). It was assumed that this uncertainty reflects the sum of all important individual uncertainties of the cumulative emissions (e.g., Sects. 3.3.1 and 4).

The experimental setup was expected to result in very similar systematic errors of the two EC systems; thus only the independent (or random) errors have to be considered for comparing the two neighboring systems (Ammann et al., 2009). As the cumulative fluxes of both EC systems were by chance of similar magnitude (Sect. 3.1 and 3.3.1), the random uncertainty of the cumulative EC fluxes was determined from the differences between the cumulative monthly EC fluxes of the two towers and resulted in a relative uncertainty of 5 % (1$\sigma$).

### 2.5.4 Footprint modeling

EC measurements yield a spatially integrated flux over a certain area represented by the flux footprint (Schmid, 2002). In the present study, this footprint typically extends over multiple grazing paddocks depending on wind direction and turbulence intensity. Therefore quantitative footprint information is needed for the comparison of the EC fluxes with the upscaled FB measurements (Sect. 2.7), and the footprint has to be checked for the spatial dimension to be sure that the measured flux is mainly dominated by the area of the system and not contaminated by the neighboring systems (either the other grazing system or fluxes originating from surrounding fields). In this study an open-source version of a backward Lagrangian stochastic dispersion footprint model (bLS) was used (Häni, 2017; Häni et al., 2018), based on Flesch et al. (2004). The flux-to-emission ratio is calculated following Eq. (2):

$$\frac{F_{EC}}{E_j} = \frac{2}{N} \sum_{i=1}^{n_j} \frac{w_{ini}^i}{w_o^i}, \tag{2}$$

where $F_{EC}$ is the measured EC flux, $E_j$ the surface emission of paddock (source area) $j$, $N$ the total number of released particles, $n_j$ the number of touchdowns within paddock $j$, $w_{ini}^i$ the vertical release velocity, and $w_o^i$ the touchdown velocity of the particles.

In order to calculate the footprint for a 30 min period, $N = 80\,000$ fluid particles were released backwards in time using the wind and turbulence parameters calculated from the sonic measurements of the EC systems. The systematic uncertainty of the bLS model was estimated to about 10 % (Flesch and Wilson, 2005; Wilson et al., 2013). The half-hourly footprint fractions of the individual paddocks were used to upscale the small-scale measurements to the EC flux footprint (Sect. 2.7) for intercomparison of the two flux measurement methods.

In addition, the seasonally integrated footprint extension was analyzed, taking into account the wind direction and $u_*$ filtering as described in Sect. 2.5.3. The analysis showed a distinct separation of the footprint distributions for the two systems (Fig. 4) with only marginal contributions of the other system (< 2.5 %). More than 80 % of the footprint contributions were from the actual rotation area (without the optional areas indicated in grey in Fig. 1a).

### 2.6 Environmental parameters

In order to relate the measured fluxes to meteorological driving parameters, an automated weather station (with data logger CR10X, Campbell Scientific Ltd., UK) was installed at the northern field next to the sonic. A WXT520 (Vaisala, Vantaa, Finland) measured the wind speed, precipitation, temperature, and barometric pressure, and global radiation was measured with a pyranometer (CNR1, Kipp&Zonen, Delft, the Netherlands).

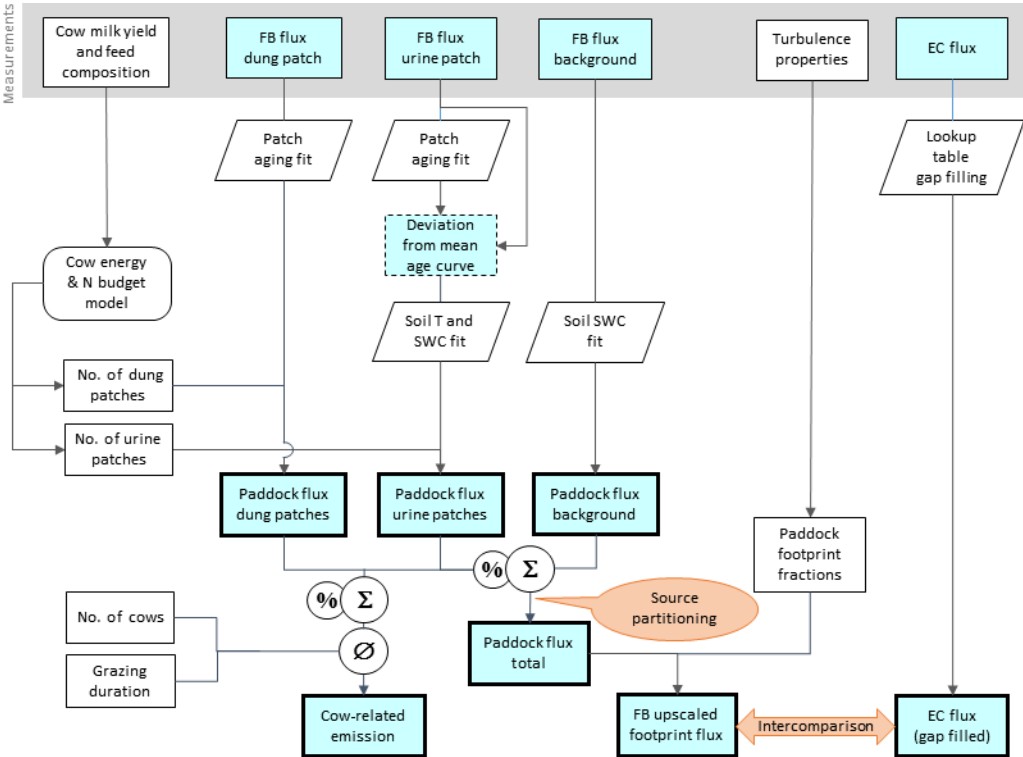

**Figure 5.** Flowchart of upscaling procedure to compare small-scale chamber fluxes with EC fluxes and to estimate the contribution of excreta emissions to the overall pasture emission. Rectangular shapes indicate time series data and other data sets. Time series data with thin frames have gaps whereas bold frames indicate complete data sets. The light blue color specifies $N_2O$ flux data. Other shapes show operations (e.g., fit or gap-filling routines).

Soil moisture and soil temperature were measured continuously with two repetitions on each pasture system close to the EC towers with ML3 ThetaProbe (Delta-T Devices Ltd., UK) devices at depths of 5, 10, 20, and 40 cm.

## 2.7 Upscaling of chamber measurements to eddy covariance footprint

Pasture $N_2O$ emissions result from a combination of "hotspot" emissions from urine and dung patches and of background emissions from the other pasture areas. Even though the FB measurements (Sect. 2.4.2) allowed for quantification of single emission sources, quantifying the contributions to the overall pasture emission is challenging due to the inherent heterogeneous nature of these emissions (e.g., spatial dimension, emission strength, temporal behavior, number of excreta patches). The EC method, in contrast, allowed us to measure the combination of all pasture sources by integrating over multiple paddocks (see footprint, Fig. 4).

FB measurements were upscaled to the EC footprint to allow a direct comparison between the two measurement approaches and to compute the contributions of the different emission sources to the overall pasture emission. The upscaling procedure is illustrated in Fig. 5. The number of urine and dung patches on the paddocks was estimated by using the daily N excretion rate (Sect. 2.3), the daily grazing duration of the cows, and a N loading of 22 g N per urination event (Misselbrook et al., 2016) and of 12.5 g N per dung pad (Cardenas et al., 2016). During the grazing season, about 12.5 dung patches $d^{-1}$ $cow^{-1}$ and a ratio of dung to urine patches of 1.3 for system M and 1.1 for system G were calculated. This compares well to values from literature (Orr et al., 2012; Oudshoorn et al., 2008; Villettaz Robichaud et al., 2011). Due to a very similar field-scale $N_2O$ emission pattern (Sect. 3.1) and comparable soil measurements (Fig. 2), it was assumed that soil parameters were homogenous on the pasture and that the soil measurements on system M were representative for the whole field.

The FB-derived $N_2O$ emissions for the different sources were analyzed for the potential driving parameters excreta age, soil temperature, and soil moisture. For this purpose various regression models (using the statistical software R; R Core Team, 2016) were tested using different predefined function types (linear, exponential, polynomial functions, sigmoidal). Based on goodness of fit and statistical significance of regression coefficients, the most suitable relationships were chosen and applied to produce continuous emission time series for the paddock areas (Fig. 5).

1. Background fluxes were parametrized as a function of soil moisture at a depth of 5 cm using the soil profile information provided in Sect. 2.6 by using a logistic regression.

2. Urine patch emissions were parametrized as an exponential decay function of excreta age. To account for different environmental conditions, the deviations of the single emissions from this temporal emission pattern were again parametrized as a function of soil temperature and moisture at a depth of 5 cm (Sect. 2.6). Upscaling fluxes to the paddock sizes involved additional information on the computed number density of urine patches (as mentioned previously).

3. Dung patch emissions were parametrized as a second-order polynomial function of excreta age. Paddock emissions were calculated by applying this function to the computed number of dung patches (as previously mentioned) per paddock.

The upscaled paddock emissions were finally compared to the EC fluxes by applying the computed footprint fractions of the paddocks (Sect. 2.5.4) in order to validate the FB measurements and to quantify the uncertainty of the upscaling process.

The area-related $N_2O$ emissions for urine and dung were also converted to emissions per cow and grazing hour. For this purpose, the upscaled paddock emissions were combined over all paddocks, accumulated for the GOP, multiplied by the pasture area of each system, and divided by the number of cows and the grazing duration (Fig. 5). The resulting emissions associated with animal excreta were then related to the excreted N of the cows (Sect. 2.3) to obtain an excreta-related EF that is comparable to the one provided by the IPCC guidelines ($EF_{3PRP,CPP}$; IPCC, 2006).

## 3 Results

### 3.1 EC fluxes

Observed EC fluxes on both pasture systems showed an almost identical temporal pattern (Fig. 6). The half-hourly fluxes on each system showed considerable variation during the grazing season with clear peaks after fertilization (grey shaded areas) and after grazing phases in the nearby paddocks (e.g., peaks in May, beginning of August). The overall highest emissions (29.0 and 24.3 g $N_2O$-N ha$^{-1}$ h$^{-1}$ for systems M and G) were measured directly after the fertilizer application, which followed a harvest of hay at the end of June. This harvest event also led to an increase in the measured $N_2O$ fluxes (0.5–3.0 g $N_2O$-N ha$^{-1}$ h$^{-1}$) which lasted less than 1 day. The partial fertilizer application in middle of August resulted in higher fluxes compared to the following one in early September. The relatively high emissions during

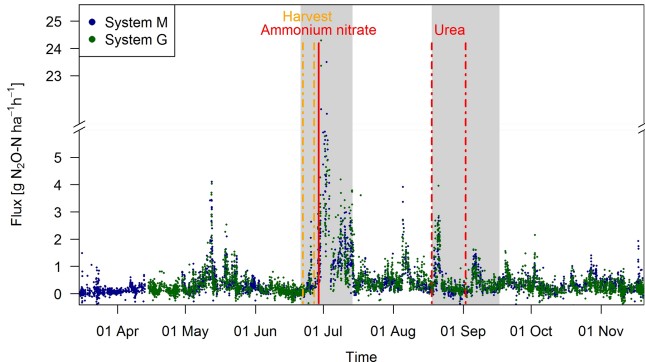

**Figure 6.** Time series of half-hourly EC flux measurements in both systems during the grazing season 2016. For the analysis of grazing-related emissions, only the non-shaded periods (GOP) were used. The vertical lines show the timings of fertilization (red) and harvest (orange) events. Dashed lines indicate that harvest and urea application were split for the western ($X.11$–$X.16$) and eastern ($X.21$–$X.25$) parts. The shaded areas indicating time periods influenced by fertilization events or harvest were excluded for the evaluation of grazing excreta-related emissions. One flux value (29.0 g $N_2O$-N ha$^{-1}$ h$^{-1}$ on system M, 30 June 2016) was skipped for better readability.

the first full grazing event at the beginning of May were characterized by high soil moisture contents (see also Fig. 2a), whereas the very wet soil conditions and the corresponding grazing break during June resulted in low fluxes in both systems. The small observed fluxes from the middle of March until the end of April resulted mainly from background fluxes and sporadic grazing (Fig. 2c). Occasional negative individual flux values between 0 and $-1.5$ g $N_2O$-N ha$^{-1}$ h$^{-1}$ were observed in both systems (7 %–8 % of the cases). However, these fluxes exclusively occurred in cases, when no defined peak in the cross-covariance function could be identified (and thus the default lag was used, Sect. 2.5.2). Thus it can be concluded that the negative fluxes were generally below the detection limit.

During the GOP (excluding the grey shaded fertilizer-influenced time periods in Fig. 6), the fluxes were still very similar for the two pasture systems M and G with a mean and standard deviation of $0.32 \pm 0.36$ and $0.33 \pm 0.37$ g $N_2O$-N ha$^{-1}$ h$^{-1}$, respectively. A mean diurnal cycle of the measured fluxes could be observed in both systems with the highest values typically occurring in the afternoon and, on average, about 10 %–20 % lower values during the night.

### 3.2 Chamber fluxes

#### 3.2.1 Comparison of pasture systems

FB chamber fluxes of background and dung patches were considerably smaller compared to the fluxes of urine patches (Table 3, Fig. 7). Freshly deposited urine patches under 3 days old could result in $N_2O$ emissions larger than 100

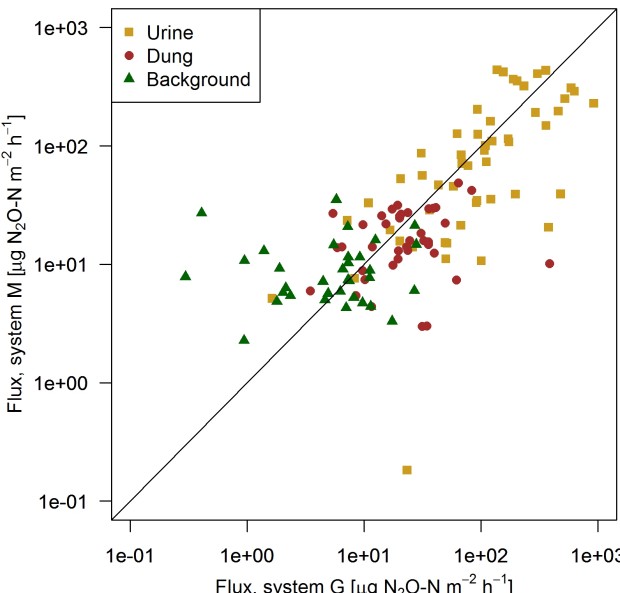

**Figure 7.** Scatterplot shows the comparison of near-simultaneous fluxes for different sources measured with the fast box on the two pasture systems. The black line indicates the 1 : 1 line.

**Table 3.** Flux measurements using the FB technique (mean $\pm$ SD) of background and excreta patches averaged over 20 days following a grazing phase. Only near-simultaneous measurements in both systems were considered.

| Measurement location | System M | System G |
|---|---|---|
| Background ($\mu$g N$_2$O-N m$^{-2}$ h$^{-1}$) | $8 \pm 8$ | $5 \pm 8$ |
| Urine ($\mu$g N$_2$O-N m$^{-2}$ h$^{-1}$) | $121 \pm 130$ | $162 \pm 190$ |
| Dung ($\mu$g N$_2$O-N m$^{-2}$ h$^{-1}$) | $16 \pm 18$ | $35 \pm 60$ |

times the values of background areas. The relative variability within the different source classes (urine, dung, background) were very high and resulted in standard deviations larger than the associated mean values. The excreta fluxes measured on system G tended to be somewhat higher in magnitude compared to system M, but no significant difference ($p > 0.05$) was found. Also, for the background fluxes no significant ($p > 0.05$) difference between the two pasture systems was observed. Therefore all FB fluxes were combined for further processing without taking into account the different pasture systems.

### 3.2.2 Dependence on excreta age

The information on the temporal pattern of the excreta and background fluxes after grazing is important for the time integration of the individual sources and for the comparison with the EC measurements. In order to analyze and parameterize the temporal evolution of the emissions, the measured FB fluxes of each source class were averaged over 3-day pe-

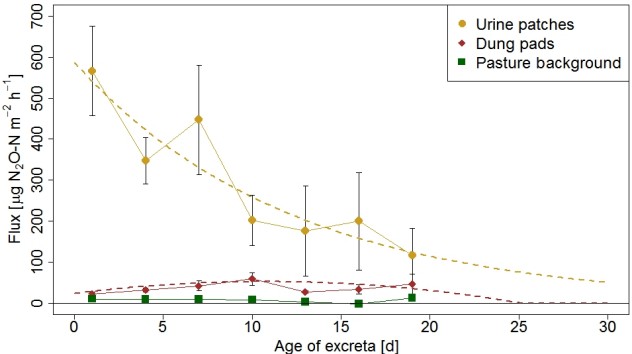

**Figure 8.** N$_2$O flux evolution with time for urine patches, dung pats, and background areas. The fluxes were measured with the fast box and averaged over 3-day periods, and the error bars show the standard error of the measurements. The standard errors for the background fluxes are smaller than the symbols. The dotted lines show the fitted curves through the averaged values of urine and dung patch emissions (see also Eqs. 3 and 4).

riods and were related to the excreta age $\Delta t_{\mathrm{EOG}}$ (Fig. 8), defined as days after EOG (Sect. 2.4.2).

Background fluxes were on average considerably smaller than excreta fluxes and showed small persisting emissions without systematic dependence on time since grazing. In contrast, for urine patch fluxes a clear relation to $\Delta t_{\mathrm{EOG}}$ was found. The highest fluxes were usually observed within the first days after the urination event. Afterwards, they rapidly decreased with time, although with a high variability that can partly be attributed to the influence of environmental conditions (see Sect. 3.2.3). The age-dependent evolution of urine patch emissions ($F_{U,\mathrm{age}}$) was parameterized with an exponential decay function fitted to the data points in Fig. 8:

$$F_{U,\mathrm{age}} = a_1 \cdot \exp^{b_1 \cdot \Delta t_{\mathrm{EOG}}}. \tag{3}$$

The coefficients of Eqs. (3)–(8) are presented in Table 4 and apply to fluxes in units of $\mu$g N$_2$O-N m$^{-2}$ h$^{-1}$.

Dung patch fluxes also showed a relation to excreta age (Fig. 8), however less pronounced compared to urine patches, and the highest emissions were typically observed between 4 and 11 days after dung deposition. However they were still smaller on average than the urine patch emissions during the entire observed age period. Because the evolution of dung emissions $F_{D,\mathrm{age}}$ after the observed 20-day age period is unclear and a meaningful functional extrapolation was not possible, we decided to use a simple second-order polynomial for parameterization purposes. This allowed us to reproduce the initial increase with age and a rapid decrease to zero beyond the measured age range: TS1

$$F_{D,\mathrm{age}} = a_2 + b_2 \cdot \Delta t_{\mathrm{EOG}} - c_2 \cdot \Delta t_{\mathrm{EOG}}^2. \tag{4}$$

The fitted polynomial function is only applicable up to $\Delta t_{\mathrm{EOG}} \approx 25$ d, where it crosses the zero line.

**Table 4.** Coefficients ($a$–$c$), corresponding indices $i$, and significance levels for the equations presented in Sect. 3.2. The equation coefficients were fitted using FB chamber measurements and yield fluxes in units of $\mu g\, N_2O$-$N\, m^{-2}\, h^{-1}$. The input quantities are the soil temperature $T_S$ (in units of degrees Celsius), time since end of grazing $\Delta t_{EOG}$ (in units of days), and volumetric water content VWC (as a dimensionless fraction).

| Equation | I | $a_i$ | $b_i$ | $c_i$ |
|---|---|---|---|---|
| Eq. (3) | 1 | 587*** | −0.082** | |
| Eq. (4) | 2 | 23* | 5.4* | −0.25* |
| Eq. (5) | 3 | 12.6*** | 0.267*** | 0.012* |
| Eq. (7) | 4 | −1490*** | 2900*** | 23.9** |
| Eq. (8) | 5 | 0.098*** | −0.086** | |

*** Significant at level $p < 0.001$. ** Significant at level $p < 0.01$.
* Significant at level $p < 0.05$.

### 3.2.3 Dependence on environmental conditions

Measured chamber fluxes were analyzed in relation to driving soil parameters (Sect. 2.6). For dung patch emissions, no relation to these parameters was found (thus $F_D = F_{D,age}$).
For background fluxes no significant dependence on soil temperature ($p < 0.05$), but a clear dependence on the volumetric water content (VWC) at a depth of 5 cm, was found. The background fluxes had a large variability and could roughly be separated by three different VWC sectors (< 0.27, 0.27–0.33, > 0.33). In the sector below a VWC of 0.27, fluxes typically ranged between −3 and 15 $\mu g\, N_2O$-$N\, m^{-2}\, h^{-1}$, whereas in the upper sector above a VWC of 0.33 the fluxes showed typical values between 0 and 30 $\mu g\, N_2O$-$N\, m^{-2}\, h^{-1}$. Nevertheless, the variability was especially pronounced in the VWC range between 0.27 and 0.33 with fluxes ranging between 0 and 40 $\mu g\, N_2O$-$N\, m^{-2}\, h^{-1}$. Thus this VWC range also comprised the overall highest background fluxes. However, averaging the fluxes by VWC intervals of 0.05 resulted in very similar values of about 12 ± 3 $\mu g\, N_2O$-$N\, m^{-2}\, h^{-1}$ above a VWC of 0.3. Hence, the measured background fluxes could be parametrized with the following functional relationship:

$$F_{BG} = \frac{a_3}{1 + \exp^{(b_3 - VWC)/c_3}}. \tag{5}$$

This logistic regression curve has a strong effect below VWC values of 0.30 but stays fairly constant at higher VWC contents and converges to a flux of 12.6 $\mu g\, N_2O$-$N\, m^{-2}\, h^{-1}$. Below a VWC of 0.2 the logistic regression converges to a background flux of 0 $\mu g\, N_2O$-$N\, m^{-2}\, h^{-1}$.

Measured urine patch emissions showed a clear response not only to the excreta age as shown in Sect. 3.2.2 but also to changes in $T_S$ and VWC. On a specific $\Delta t_{EOG}$, $F_{U,age}$ could vary significantly and typically correlated with soil conditions. The highest flux (5117 $\mu g\, N_2O$-$N\, m^{-2}\, h^{-1}$, $\Delta t_{EOG}$ = 6 d) was measured at a $T_S$ of 18 °C and a VWC of 0.42 while the lowest measured flux (34 $\mu g\, N_2O$-$N\, m^{-2}\, h^{-1}$) on

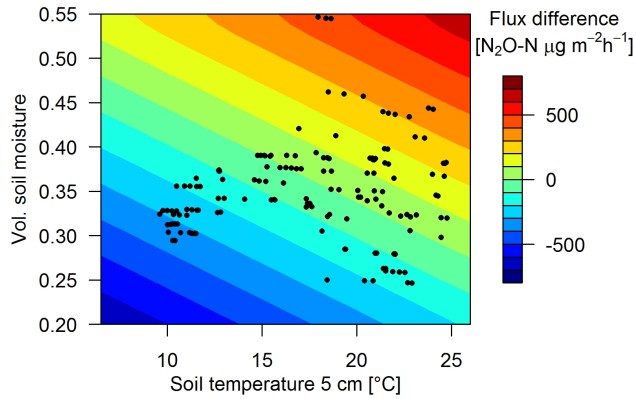

**Figure 9.** Surface plot shows the estimated $N_2O$ flux deviation (Eqs. 6, 7; $Corr_{U,env} = 0$) from the exponential fit (Eq. 3) for urine patches depending on soil moisture $VWC_U$ and temperature at a depth of 5 cm. The black dots indicate the conditions under which flux measurements with the FB were obtained.

a similar $\Delta t_{EOG}$ was measured at a low $T_S$ (1 °C) and a lower VWC (0.3). Maximum positive measured FB flux deviations (Sect. 2.7) from Eq. (3) were generally observed for wet (VWC > 0.45) and warm (> 17 °C) soil conditions while low $T_S$ and VWC resulted in negative flux deviations. Thus, the final regression model for urine patch emissions (Eq. 6) consists of multiple equations (Eqs. 3, 7, 8) which relate the measured fluxes to the temporal decay (Eq. 3) and a deviation $\Delta F_{U,env}$ to this decay, where $\Delta F_{U,env}$ was parametrized as a function of environmental driving parameters $T_S$ and $VWC_U$ (Eqs. 7 and 8, Fig. 9).

$$F_U = F_{U,age} + \Delta F_{U,env} \tag{6}$$

$$\Delta F_{U,env} = (a_4 + b_4 \cdot VWC_U + c_4 \cdot T_S)$$
$$\cdot Corr_{U,env} (\Delta t_{EOG}) \tag{7}$$

$Corr_{U,env}$ corrects $\Delta F_{U,env}$ for different urine patch ages as the deviation can be larger for relatively new patches compared to older ones. This correction factor was found to be a linear relationship ($p < 0.01$) between 1.35 for a $\Delta t_{EOG}$ of 0 days (after the patch deposition) and 0.35 after 20 days. $VWC_U$ (Eq. 8) accounts for different soil moisture conditions at the surface below a urine patch and nearby background areas and was parametrized as a function of background VWC and $\Delta t_{EOG}$ (Eq. 8).

$$VWC_U = VWC + a_5 \cdot \exp^{b_5 \cdot \Delta t_{EOG}} \tag{8}$$

## 3.3 Upscaled chamber fluxes

### 3.3.1 Comparison between upscaled chamber and EC fluxes

Generally the field-scale fluxes represent the area integral of management-related (excreta patches) and environmen-

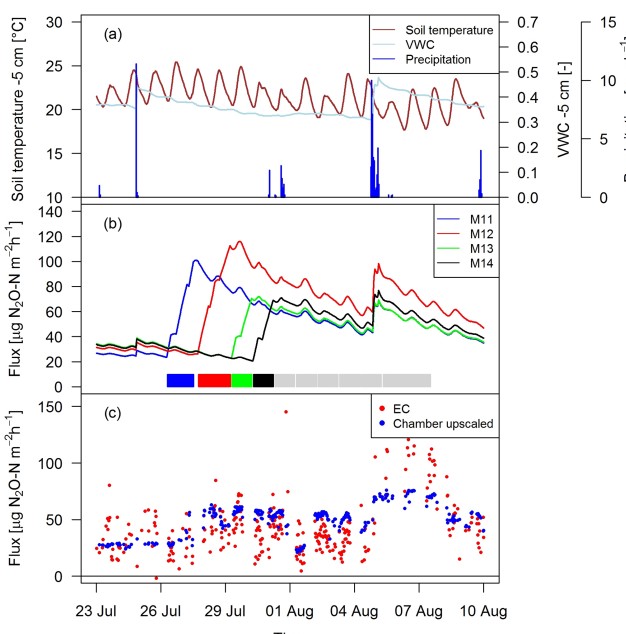

**Figure 10.** Time series of **(a)** environmental parameters and **(b)** upscaled FB fluxes (Sect. 2.7) for different paddocks (M11–M14) in system M. The colored rectangles at the bottom show the grazing phases on the four considered paddocks (grey colors indicating grazing on the remaining paddocks). **(c)** N$_2$O fluxes by EC and upscaled FB during a full rotation between 23 September and 10 August 2016 for system M.

tally driven small-scale fluxes. Therefore the relationships presented in Sect. 3.2.2 (dependency on excreta age) and Sect. 3.2.3 (environmental driving parameter) were applied to upscale the FB measurements to the paddock size during the GOP.

As shown as an example in Fig. 10 for an 18-day period, the magnitude of the management-related upscaled paddock fluxes depended mainly on the grazing duration on the single paddocks (similar slope for different paddocks M11–M14). The maximum of the emissions was typically calculated at the end of the grazing period on the respective paddocks. The lower limit of the fluxes was given by the estimated background fluxes, especially at the beginning of a new rotation, and stayed therefore rather constant for VWC values above 0.3 (Eq. 5, Sect. 3.2.3). Variations in environmental conditions (mainly important for soil moisture) led to rapid changes in the emission level as long as significant urine patch emissions were present. These rapid variations typically occurred after stronger precipitation events (as shown in Fig. 10a for on-site meteorological and soil measurements).

Upscaling the paddock fluxes to the EC footprint allowed a direct comparison with the EC fluxes on a half-hourly basis (Fig. 10c). The upscaled FB fluxes compared well in magnitude with the measured EC fluxes and showed a similar temporal behavior. While generally a response to variations

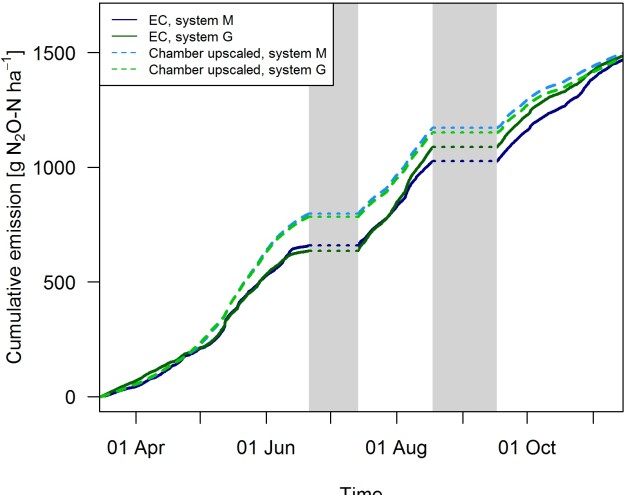

**Figure 11.** Cumulative emissions for both systems obtained with the FB and EC technique during GOP 2016. The grey shaded bars indicate time periods which were excluded due to significant overlapping N$_2$O emissions from fertilization/harvest and grazing (Sects. 2.5, 3.1).

in environmental driving parameter could be observed, it was less pronounced for the upscaled FB fluxes in comparison to the EC fluxes.

Gap filling of the EC fluxes (Sect. 2.5.3) allowed the calculation of the cumulative N$_2$O emissions during the GOP (solid lines in Fig. 11). These area-related emissions were very similar between the two systems throughout the GOP with seasonal sums close to 1500 g N$_2$O-N ha$^{-1}$. Cumulating the N$_2$O emissions not only enabled a more quantitative comparison between the systems, but also allowed a better comparison between the two measurement approaches (Fig. 11). The emissions of the upscaled FB matched the EC emissions rather well with differences of the seasonal sums below 3 %. Distinct differences were mainly observed in May and June when FB-derived emissions were significantly overestimated compared to EC. At the end of the grazing period and averaged over both systems, slightly higher emissions were estimated from the upscaling routine compared to the measured EC emissions. Monthly absolute differences between the cumulative EC and the upscaled cumulative FB sums were normally distributed ($p < 0.05$) with $1\sigma$ values of 26 % and 25 % for systems M and G, respectively. Within this uncertainty range no difference between the two measurement approaches was observable.

### 3.3.2 Emission breakdown into contribution sources

The excellent match between the EC fluxes and the upscaled chamber-based fluxes showed that the applied relationships which used excreta age and environmental parameters as input (see Sect. 3.2) were reasonable and allowed the separation into single emission sources (Fig. 12). Except for the be-

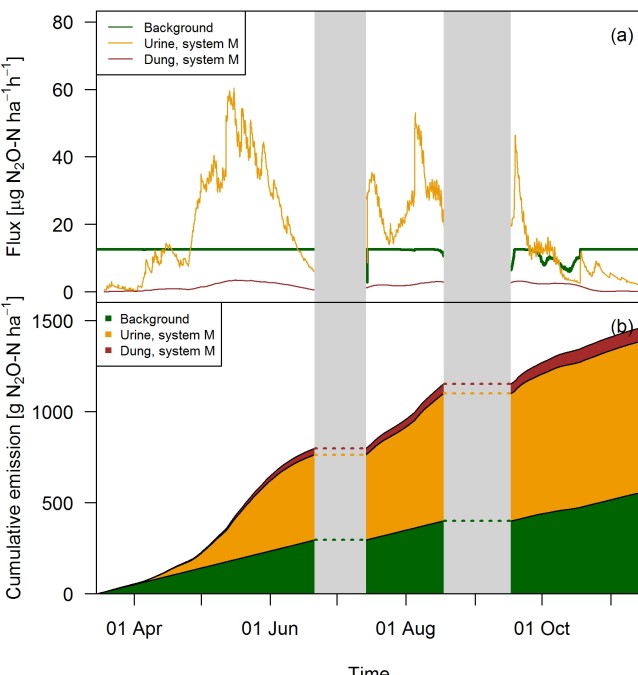

**Figure 12. (a)** Time series of upscaled FB fluxes averaged over all paddocks of system M for all three emission sources during the grazing season 2016, and **(b)** retrieved cumulative emission contribution of the emission sources to the overall field emission.

ginning of the grazing season when grazing rate was very low (see Fig. 2), the urine patch emissions dominated the field-scale fluxes. In May, this effect was even more pronounced due to the wet soil conditions. Based on the upscaling, the

5 averaged urine patch emissions of both systems were responsible for about 57 % of the pasture emissions. Background contributed to about 38 % and dung emissions to about 5 % to the overall field emissions. Both systems had very similar contributions, with only 1 % difference in the dung contribu-

10 tion as a result of a different N excretion ha$^{-1}$ on the pasture by dung (Table 5). Background emissions were simulated to be constant for most of the GOP due to the weak sensitivity of Eq. (5) to VWC and the undetected sensitivity towards soil temperature.

15 **4 Discussion**

**4.1 Area-related and animal-related emissions**

The EC and upscaled FB emission results presented in Sect. 3.3.1 are normalized by area and showed the emissions for the EC footprint (see also summary in Table 5). The good

20 agreement with a relative difference below 1.5 % for yearly sums (which is far below the uncertainty range; see Table 5) between the two independent approaches supports their quality (including the upscaling procedure) in this study. We assume that the EC fluxes are on average representative for the

whole pasture system, although the contribution of the cen-25 tral paddocks $X.11$, $X.12$ and $X.21$, $X.22$ to the EC foot-print is generally higher than the contribution of the other more distant paddocks (Fig. 4). We found no indication of significant differences between the paddocks concerning soil conditions, vegetation productivity, or other characteristics 30 (data not shown). An alternative upscaling of the FB measurements to the entire pasture system (without taking the EC footprint into account) representing the average emission over all rotation paddocks (Table 5, FB emissions upscaled to pasture system) differed less than 4 % from the EC footprint-35 related emissions.

For assessing the effect of the N-reduced diet on excreta-related $N_2O$ emissions, the emissions per cow and grazing hour were compared, taking into account the different pasture sizes for systems M and G (according to Sect. 2.7). The cor-40 responding results in Table 5 show about 25 % lower excreta-related $N_2O$ emissions per cow for the herd in system M than for the herd in system G during the GOP. The difference is not statistically significant, probably due to the considerable uncertainties resulting from the FB upscaling procedure. 45 For comparing the two herds the parallel direct EC measurements are better suited as only random uncertainties have to be taken into account (Sect. 2.5.3), yet they also include the background emissions. The EC-based $N_2O$ emissions per cow were $0.20 \pm 0.03$ and $0.27 \pm 0.05$ g $N_2O$-N cow$^{-1}$ h$^{-1}$ for 50 system M and system G, respectively, and resulted in a significant difference of $0.07 \pm 0.02$ g $N_2O$-N cow$^{-1}$ h$^{-1}$ between the two herds. This indicates the ability of a N-adjusted forage to reduce the excreta N content and related emissions of $N_2O$. It has to be noted that this evaluation does not comprise 55 the full $N_2O$ emission of the pasture fields or of the milk production system but only the emissions related to grazing excreta following the IPCC concept (EF$_{3PRP,CPP}$; IPCC, 2006). Any further $N_2O$ emissions, e.g., related to fertilizer application on the pastures or the supplement maize production, 60 were not taken into account here. A comparison of entire production systems would require many additional assumptions outside the specific scope of this study. It also has to be considered that the N optimization of the diet is not necessarily linked to the supplemental feed of arable crops like maize 65 but may also be achieved with different feed strategies (e.g., grass varieties with a high content of water-soluble carbohydrates; Misselbrook et al., 2013).

**4.2 Excreta-related emission factor**

Area- or cow-related emissions as described in Sect. 4.1 70 enabled the comparison of the different measurement approaches and the discussion of the diet effects on $N_2O$ emission. However, results presented in literature or used in national inventories typically relate emissions to the N inputs within a given time period using EFs. The annual excreta-75 related EF (Table 5) in the present study was $0.74 \pm 0.26$ % for system M and $0.83 \pm 0.29$ % for system G. These EFs are

**Table 5.** Summary of cumulated grazing-related emissions for both pasture systems during the GOP 2016. The table shows the emissions per area (first part), pasture area, excreta N input (second part), and the emissions per cow and grazing hour as well as the calculated EFs (third part). The uncertainties are given as $1\sigma$.

| Parameter | System M | System G |
|---|---|---|
| EC emission (kg $N_2$O-N ha$^{-1}$) | $1.50 \pm 0.21$ | $1.51 \pm 0.27$ |
| FB emissions upscaled to EC footprint (kg $N_2$O-N ha$^{-1}$) | $1.51 \pm 0.39$ | $1.50 \pm 0.37$ |
| FB emissions upscaled to pasture system (kg $N_2$O-N ha$^{-1}$)[a] | $1.48 \pm 0.38$ | $1.48 \pm 0.37$ |
| Pasture system area (ha) | 1.88 | 2.51 |
| Excreta N total (g N cow$^{-1}$ h$^{-1}$) | $16.5 \pm 1.2$ | $19.6 \pm 1.5$ |
| Urine N (g N cow$^{-1}$ h$^{-1}$) | $10.2 \pm 1.1$ | $13.0 \pm 1.3$ |
| Dung N (g N cow$^{-1}$ h$^{-1}$) | $6.3 \pm 0.7$ | $6.5 \pm 0.7$ |
| FB excreta emissions (g $N_2$O-N cow$^{-1}$ h$^{-1}$) | $0.12 \pm 0.04$ | $0.16 \pm 0.05$ |
| EF excreta total (%)[b] | $0.74 \pm 0.26$ | $0.83 \pm 0.29$ |
| EF urine (%)[b] | $1.09 \pm 0.43$ | $1.15 \pm 0.43$ |
| EF dung (%)[b] | $0.16 \pm 0.06$ | $0.17 \pm 0.06$ |

[a] Average emissions over all paddocks. [b] Based on FB emissions upscaled to the pasture system.

based on the combined, upscaled FB measurements of urine and dung patches (see Sect. 3.3.2) relative to the N excreted on the pastures during the GOP (Table 5). Their uncertainty is defined by the combined uncertainty of the upscaling method (Sect. 3.3.1) and the N input estimation (7.5 %). The difference in the EFs between the systems is therefore not statistically significant.

The resulting EFs were significantly smaller compared to the proposed default $EF_{3PRP,CPP}$ of the IPCC guidelines for cattle excreta (2 %; IPCC, 2006), which makes the use of the latter in the Swiss national inventory questionable. The upscaled FB measurements also allowed us to separately calculate the EFs for urine and dung. We found EFs of $1.12 \pm 0.43$ % and $0.16 \pm 0.06$ % for urine and dung, respectively (average of both systems due to small difference; see Table 5). These EFs are comparable to the results of newer studies (0.59 % and 0.26 % for urine and dung patches combined from cattle and sheep; Cai and Akiyama, 2016; 1.18 % and 0.31 % for cattle urine and dung; Krol et al., 2016). The large difference of the EFs for urine and dung also supports the suggestion of Krol et al. (2016) to disaggregate the EF by excreta type in emission inventories. The implementation of excreta-specific EFs could allow for a more precise calculation of the grazing-related $N_2$O emissions, e.g., as dietary effects regarding the N intake predominantly affect the excreted urine N, which is the main source for the high $N_2$O emission associated with excreta (Dijkstra et al., 2013).

The background emissions measured by FB cannot be attributed to a specific N input in a quantitative way, but the annual sum of 1.03 kg $N_2$O-N ha$^{-1}$ yr$^{-1}$ from this study (extrapolated using Eq. 5 and VWC data for the whole year) compares well with background emissions reported by a meta-study of Kim et al. (2013, median 0.7 and mean 1.52 kg $N_2$O-N ha$^{-1}$ yr$^{-1}$) for agricultural lands. In agricul-

tural systems, background emissions are usually determined as emissions from (managed) plots receiving no fertilization in the study year. Thus they still include N inputs from plant residues and atmospheric deposition. Background emissions are also often regarded as a late effect of fertilization events from previous years (Bouwman, 1996; Gu et al., 2009). On pastures, background emissions may additionally result from trampling of the cows that can further stimulate the $N_2$O production via denitrification due to soil compaction (Bhandral et al., 2007).

## 4.3 Upscaling of FB fluxes

Urine patch emissions were parametrized with an exponential decay and maximum initial emissions of about 600 µg $N_2$O-N m$^{-2}$ h$^{-1}$ that is close to the maximum averaged emissions measured by Barneze et al. (2015) from manually applied urine in laboratory conditions and on a grassland. A strong emission response to urine application was generally reported in the literature, however, with a large range of different emission dynamics and magnitudes (e.g., two emission peaks due to nitrification and denitrification, emission peak after a few days with near exponential decay afterwards, significant emissions after weeks to a month; Bell et al., 2015; Cardenas et al., 2016; Chadwick et al., 2018). Similar to our study, reported dung patch emissions by those studies were much lower compared to urine-induced emissions.

We found that pasture emissions were dominated by excreta-related emissions during the GOP (about 60 %). On a seasonal basis, the upscaled aggregated fluxes compared well with the gap-filled EC measurements, which also indicates the validity of the source attribution in the upscaled emissions. Especially during time periods when both FB fluxes

and EC fluxes were measured (July–October), the agreement between the systems was very good.

However, the parameterizations used for upscaling resulted in a poor performance for certain soil conditions. The limited sensitivity towards changes in VWC of the background fluxes is probably due to the fact that FB measurements were mainly performed during dry soil conditions. We have no explanation why we did not find a significant sensitivity of the background fluxes towards changes in $T_s$ TS2 as reported by other studies (Butterbach-Bahl et al., 2013; Schindlbacher et al., 2004). Typically, increasing soil temperature leads to increased soil respiration, which subsequently can lead to a depletion of soil oxygen and further to higher denitrification rates. In contrast to background fluxes, the urine patch emissions showed a clear response to changes in $T_s$ TS3 and VWC. This effect could be parametrized with a bilinear regression (Eqs. 7 and 8). This regression led to high upscaled emissions from urine patches, especially during wet soil conditions, and subsequently to an overestimation of the cumulative emissions in May and June compared to the EC systems. $N_2O$ emissions often have an emission maximum during moderately wet soil conditions (VWC between 0.40 and 0.45) while completely anaerobic conditions at saturated VWC can lead to a complete denitrification with only marginal $N_2O$ emissions (Butterbach-Bahl et al., 2013). Such conditions have been very rare during the FB measurements (see Fig. 9) and therefore may not be adequately represented in the derived parameterization. A general trend towards lower emissions during very wet soil conditions was also observed by the EC systems (not shown). However, in order to avoid mixing results of the different measurement systems and thus reducing the explanatory power of the system intercomparison, we decided to base the environmental regression analysis (Sect. 3.2.3) only on data measured by the FB.

### 4.4 Advantages and problems of experimental setup

The presented field campaign was designed to estimate the $N_2O$ emissions of two parallel grazing systems and to compare different feeding diets of the herds. Field-scale emissions derived by the EC method resulted in a wide range of measured emissions, which were mainly driven by environmental and management-related parameters. Nevertheless, the setup with two towers allowed for a good comparison with a sufficient number of measured fluxes from both systems. Due to a delayed installation of the EC tower at system G, all fluxes prior to the middle of April had to be gap filled, which resulted in a higher associated uncertainty.

The excreta N input derived by the animal budget approach at a temporal resolution of 1 day was needed in order to quantify the EF of the two systems and to upscale FB chamber measurements to the field scale. Nevertheless, direct measurements would have been preferable. However, as the N content in the excreta is highly variable (Betteridge et al.,

2013) on seasonal (e.g., due to variability in the N content of the fodder) and short-term scales (e.g., different urine volume, different cows, difference between day and night), continuous measurements throughout the grazing period for a representative number of cows would have been needed. This is only possible with measurement equipment directly placed on the cow. However, these measurements are often limited regarding animal welfare, are not well established and the associated uncertainty is still considerable (Misselbrook et al., 2016). Thus, they were not used in this study.

The combined approach of EC and FB measurements allowed the quantification of the uncertainty of the upscaling routine, and the good match between the two measurement approaches also validates the resulting contributions of the different emission sources on the field scale. The uncertainty associated with the upscaling mainly resulted from missing FB measurements during wet soil conditions (e.g., in spring), which prevented the use of a more complex parameterization of environmental driver effects on background and urine emission. In summary, the experimental setup resulted in robust field-scale emissions, allowed us to compare the two pasture systems, and yielded source-specific emission factors for dung and urine patches.

### 5 Concluding remarks

The temporal dynamics of background areas and excreta patches were observed by fast-box (FB) chamber measurements on the pasture. We found no significant temporal pattern of the background fluxes. Urine patch emissions were parametrized by an exponential decay with time, whereas a less pronounced dependency on excreta age of dung emissions was observed. This relation was parametrized with a quadratic function and a maximum after about 10 days. On a field-scale level, urine patch emissions dominated the pasture emissions during the grazing season. Nevertheless, background fluxes contributed significantly to the pasture emissions as well. The origin of these background fluxes is still uncertain and should be addressed in further studies.

The combined approach with EC and FB measurements proved to be appropriate to observe and quantify the magnitude of the pasture emissions and to calculate the contribution of the single emission sources. The different diet of the cows resulted in an excreta-related $N_2O$ emission difference of about 25 % between the two cow herds and revealed the large potential of a N-optimized feeding strategy to reduce grazing-related $N_2O$ emissions. In this study, the N optimization was achieved by adding maize silage to the fodder in system M. However, a reduction in excreted N can potentially be realized by other means as well (e.g., grass varieties with a high content of water-soluble carbohydrates). The excreta-related EFs derived from the upscaled FB measurements were $0.74 \pm 0.26$ % for system M and $0.83 \pm 0.29$ % for system G and were thus significantly lower compared

to the current default EF of 2 % for cattle excreta provided by the guidelines of the IPCC. The findings also exhibited clear differences in the individual EFs for urine and dung (1.12±0.43 % and 0.16±0.06 %, respectively, averaged over systems M and G), suggesting a corresponding disaggregation in emission inventories.

*Data availability.* Data obtained in this study are available online at https://doi.org/10.5281/zenodo.2601821 (Voglmeier et al., 2019).

*Competing interests.* The authors declare that they have no conflict of interest.

*Acknowledgements.* We gratefully acknowledge the funding from the Swiss National Science Foundation (project NICEGRAS, no. 155964) and from the Swiss Federal Office for the Environment (contract no. 16.0030.KP/P031-1675). We wish to thank Lukas Eggerschwiler, Robin Giger, Walter Glauser, Andreas Münger, and Jens Leifeld for support in the field and helpful discussions. We especially acknowledge the contribution of Harald Menzi in the design and planning of the experiment and Arjan Hensen for lending us the fast box and advising how to use it. We are grateful to Albrecht Neftel for the helpful discussions and advice concerning the measurements. We thank Daniel Bretscher for the support with the N balance computation of the cows and the discussions of these data. Karl Voglmeier was additionally supported by a MICMoR fellowship through KIT/IMK-IFU.

*Review statement.* This paper was edited by Andreas Ibrom and reviewed by two anonymous referees.

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

## Remarks from the typesetter