# Peer review of "Grazing related nitrous oxide emissions: from patch scale to field scale"

_Biogeosciences, 2018_

## Referee Comment (RC1) · Anonymous Referee #1 · 26 Nov 2018

The study presents N2O emission measurements of two grazing systems, which differed in the energy to protein balance of the cows' diet. Measurements were carried out by two independent approaches, eddy covariance and fast box technique, and the small scale fluxes (measured with fast box technique) were up-scaled to match EC flux footprints for comparison. The good match between the two approaches justified the disaggregation of the different emission sources (urine, dung, background) on the pasture. This enabled the authors to present EFs for total, urine and dung fluxes, indicating the need to disaggregate grazing related EFs by excreta type. This information is very valid for policy makers as well as for the scientific community. The methodologies used in the manuscript are scientifically sound and the manuscript is well structured. However, one main point that needs to be addressed is the final conclusion; I strongly

disagree that you can conclude that an optimised diet (with additional maize silage) in system M leads to a 25% reduction effect for N2O emissions based on the smaller area needed for grazing. The authors need to take the N2O emissions related to the maize production used in the diet into account; otherwise this comparison is not valid. Generally the authors need to be more careful with figure and table captions. The structure of some tables needs to be improved and the authors need to be more carful with units, especially when presenting cumulative fluxes (table 5). There are many abbreviations that were not explained (e.g. ECM, FAD, Q, A, V) or not very clearly (FD, FU, FU,temp, Fbg), which makes equations difficult to understand (section 3.2.2 and 3.2.3). Improving figures and tables and explaining abbreviations will help to make the manuscript easier to read. I recommend this manuscript for publication in bg after thorough consideration of all the comments listed below.

Detailed comments:

Abstract:

P1, line 11 change "measurement of" to "measurement from"

P1, line 12 I suggest to change "The diet for the cows.." to "The diet for the two herds of cows"

P1, line 14 Change "animal budget model" to "animal nitrogen budget model"

P1, line 18 Replace "to the field" with "to field level"

P1, line 21 Add "due to the required greater area for grazing" at the end of the sentence. Please see general comment.

P1, line 25 insert "respectively" after "for urine and dung"

P1, line 26/27 This conclusion is not correct, as the N2O produced during the production of Maize fed to the animal is not included in the calculations!

Introduction:

P1, line 30 Please add a reference for the GWP of N2O.

P2, line6 Please insert "from excreta" after "N loading... was shown previously" and insert N after exceptionally high.."

P2, line16 Please give a suggestion of how emissions could be reduced if individual contributions are better understood.

P2, line31 I suggest to change "..different systems (intensive-extensive, different crops, land / lake,...)" to "..different systems (e.g. intensive vs extensive, different crops, land vs lake)"

Material and Methods:

P3, line 15 use average values for clay, silt and sand from table 1

P3, line 17 Change "vegetation consists" to "vegetation consisted"

P3, line 17 The range of 10-50 and 7-40 % of Lolium and Trifolium is quite large; could you give an average $\pm$ stdev and the method of how it was assessed?

P3, line 19 I suggest to change the last sentence to: " Beside the N input through excreta from the animals, N was applied through fertiliser at a rate of 120 kg N ha1 per year between 2007 and 2015."

P3, line25 You write that the optimized protein content reduced the N input to the pasture. Did you measure this? Otherwise, just write that it was expected to reduce the N input to the pasture.

P3, line26/26 Please add "(see table 2)" at the end of the sentence

P4, line 7 Add comma after "soil conditions"

P4, line 18 Please add the name of the model and give a reference

P4, line18 Change " In short, this model balances the N flows..." to "The model balances the N flow..."

[Figure]

P4, line18/20 Please add "(see table 2)" at the end of the sentence

P4, line20 I suggest to change this sentence to "For further details see Voglmeier et al (2018), where the uncertainty of the total N and urine/dung N was estimated to be 15 % ..."

P4, line20 Move this sentence up to the end of the sentence (line18-20)

P4, line27 add "respectively" after "X.21.."

P5, line3 change "blue areas in Fig. 3a)" to "dark blue areas in Fig. 3a"

P5, line8 change "..with the fast-box..." to "..with a fast-box..." to

P5, line10 change (Fig.2) to (Fig.2b)

P5, line11 insert "the" after "days after.."

P5, line17 replace "The gas.." with "The air.."

P5, line20-21 I am not quite sure if I understand this modification. Did you add a vent to the box? Then better to call it vent than inlet as the inlet is connected to the QCL. Please be more specific: I assume the 4 cm is the diameter and the 1m is the length of the vent tube and the 10 cm is the length of the foam material within the tube? What is the foam made of?

P5, l22 As you don't show or discuss any soil respiration measurements, I suggest to delete this information.

P5, line22/23 Insert "N2O" after "The increase in..."

P5, line24 Insert "the" after "The inflow off..." I assume by "the inlet" you mean " the vent"? As the FB chamber is a closed dynamic system (acc. to Hensen et al.2006).

P5 line26 please add explanations to all abbreviations used in equation 1 (V, A, Fcham)

P6 line12 please give the make of the thermocouple

P6 line21-22 I suggest to delete "because sometimes. . . .for further processing." and replace with ". . .measurements (see also Fi. 3b ), in order to exclude possible old urine patches (of previous management rotations)."

P7 line13 Instead of "separated" I suggest to write "The distance of the inlets of the QCL from the centre of the sonic head were around 20 cm. . ."

P7 line14 Replace "some 20 meters" with " about 20 meters"

P7 line18 I suggest to change "the measurements and fluxes of an online flux calculation" to " the N2O concentrations and fluxes, calculate with an on line flux calculation,. . ."

P7 line24 500; add unit

P7 line25 Move the reference after "technique"

P8 line1 insert "was used" after " . . .the default lag."

P8 line4 change "..can lead to significant.." to "..can lead to a significant. . ."

P8 line12 change (see Fig 2) to (see Fig. 2c). Please show the harvest event in the Figure (see comments to Figures)

P8 line23 Change "..due to low u* (<0.07 ms-1)" to ". . .due to low friction velocity (u*, <0.07 ms-1)

P8 line24 change "(-2°-6°)" to "(-2° to 6°)"

P8 line27 Change "The data gaps had a diurnal pattern.." with " The occurrence of data gaps showed a diurnal pattern. . ." and join the two sentences together ". . .during the night, which was driven by. . ."

P8 line 33 Can you give a time period for the soil temperature classes?

P9 line16 change "(either the other system.." to "(either the other grazing system.."

P9 line18 insert (bLS) after "footprint model"

P10 line20 Delete "number" before "ratio"

P10 line23 insert "of N2O" after "measurements"

P10 line25 change "were analysed for potential driving parameters (excreta age, soil temperature, soil moisture)." To "were analysed for the potential driving parameters excreta age, soil temperature and soil moisture."

P11 line8/9 This last sentence is not clear. Please clarify which fluxes you are talking about (individual emission source; paddock or system M/G?)

Results:

P11 line12 "they varied significantly"; significantly different from what? Background fluxes?

P11 line13 mention the harvest, were they increased after the harvest?

P12 line14 change "were related to the days after EOG (defined as excreta age, Sect. 2.4.2) deltatEOG" to "were related to the excreta age deltatEOG (defined as days after end of grazing (EOG), Sect. 2.4.2)"

P12 line15-19 In Figure 8 FU,temp and FD,temp are fluxes averaged over 3 days, while in the text you are describing average daily values (?), which is confusion. I suggest to show average daily values in Figure 8, or use different abbreviations (e.g. FU,temp3d vs FU,temp1d), or only discuss 3d averages in the text. Its not clear what the "absolute highest FUtemp" is (5117 ug N2O-N m-2h-1), if the highest average value is 660 ug N2O-Nm-2h-1.

P12 line20 you mention that Dung related emissions "showed a relation to excreta age", please mention what kind of relation. Please change "dung patch emissions" to "dung patch fluxes"

P12 line25 Were the background fluxes not also sign. smaller compared to dung patch emissions? Looks like it in figure 8.

P12 line26 I suggest to change ". . .to the averaged measured flux values within 3 days." to "to the measured flux values average over 3 days"

P12 line32 How do you justify to set negative values to zero?

P13 line6 As "FD,temp" is not influenced by environmental conditions it equals FD (?)This should be stated here.

P13 line6 Please move "was found" to the end of the sentence ending on line7.

P13 line7-14 Please define the three sectors. I suggest to insert (<0.27, 27-33, >0.33) after ". . .by three different VWC sectors." It would help to show this in a graph.

P13 line13 do you mean "similar values" or comparable to what? Can you add a stdev?

P14, line17 It's not clear where the grazing period ends, therefore please add this information into the table (see comments table 10b).

P14 line25-27 these two sentences are not very clear. What do you mean by variations? The magnitude of fluxes varied less? I suggest to replace "rather limited" with "less pronounced"

P14 line29 change "emission" to "emissions". I suggest to replace "comparable" with "similar"

P15 line2 But in Figure 11 it looks like fluxes were slightly higher for up-scaled FB fluxes.

Discussion:

P15, line15 delete brackets around "manually applied".

P15, line20/21 I don't understand this sentence

P15, line25 Include "(data not shown)" at the end of "other characteristics" as you didn't show any productivity (yield). Delete "Also" at the beginning of the next sentence.

P15, line 26 Change "(FB system emissions)" to "FB up-scaled to system emissions)"

P15, line 28-p16 line 4 This paragraph is difficult to understand. I strongly disagree that you can conclude that an optimised diet in system M leads to a 25% reduction effect for N2O emissions based on the smaller area needed for grazing. You need to take the N2O emissions related to the maize production into account, otherwise this comparison is not valid.

P16, line 18/19 This sentence is not clear. Change to "Hence the EFs for excreta retrieved from manually applied urine and dung patches and those retrieved by the EC method"

P16, line 23 1.03 kg N2O-Nha-1y-1, please explain in more detail how this value was calculated. This value should have been shown in the results section 3.3.2.

P16, line 26 Change "events in previous" to "events from previous"

P16, line 26/27 This last sentence is out of context

P17, line4 delete brackets around "or urine patch emissions".

P17, line16 Delete "additionally" and "certain"

P17, line21 Insert "upscaled" after "..led to very high"

P19, line11 Insert "us" after "..for lending"

References:

Change order of Flesch 2005 and Flesch 2004

Tables:

Table 1: As soil depth is not really a parameter I suggest to re-arrange the table;

one column for each soil depth, with missing values in each column as the different parameters have not been measured in all soil layers

Table 2: What does ECM stand for? Please explain abbreviation (maybe in footnote). Have the animals been weighted before and after the experiment? Was the weight increase considered in the calculations of the excretion (heavier animals will excrete more)? Table capture: I suggest to change "and resulting urine N and faeces N of the Swiss dairy …." To " and resulting modelled urine and faces N using the Swiss dairy…"

Table 3: Please add information of flux measurement method (FB)

Table 4: Please describe what the different equations are: Parameterisations of 3 day average fluxes from EB measurements, split into background, dung and urine fluxes.

Table 5: I assume that you are showing cumulative fluxes. You need to mention this together with the time scale (per GOP?). I suggest to simplify the table by only having two parts; add a dotted line above the N input and to move the FB urine and FB dung fluxes above the EFs. Please add N input from dung. What does FAD stand for? What is "EC integral system emission EC," ? Reading in the text (P15, line 28-31) I have the impression these fluxes are up-scaled FB fluxes to the whole system. If they are EC emissions, please describe more carefully in the text. The unit is confusing as it is an emission (concentration per area per time). Reading in the discussion I understand what you mean, but in the table it's not clear. Maybe you can explain in a foot note. "EF total" is calculated from EC, while "EF urine" and "EF dung" are calculated from up-scaled FB measurements (or not?) This needs to be stated clearly.

Figures:

Figure 1: P29 line4 Insert "(triangles)" after "the two EC towers…"

Figure 2 b): Move the legend, or change the scale so the bars for the high precipitation events in June and July are not cut off.

Figure 2d): Add arrows for fertiliser application dates and for harvest date

Figure 3: Change the area showing a) to being transparent

Figure 5: Please explain the reason for dotted frame. It's confusing that the lines connecting #dung and # urine patches to "Paddock flux dung patches" and "paddock flux urine patches" cross the arrows leading to "paddock flux background" and "paddock flux urine patches", it looks like they are feeding into them as well. Try to show clearer (maybe with a curved line over the crossing line).

Figure 6: Add arrows for exact fertilisation and harvest events. Add the date of the skipped value. It would be good to include the information of grazing periods in the graph to explain the increased fluxes.

Figure 7: I suggest to change the unit to ug N2O-N m-2h-1, it makes it easier to read the values in the graph and to compare to values you describe in the text (for Figure8, where ug N2O-N m-1h-1 are used). In the legend insert "from different sources" after "the comparison of fluxes.." and add the information that the fluxes were measured with FB technique.

Figure 8: Same comments about units as for Fig. 7. Please add in legend that fluxes were measured with FB. Are there any standard errors for the background fluxes, or were they too small to be seen? I suggest to change to x-axis description to "age of excreta [d]". Give information about the fitted curves (refer to equation 3+4).

Figure 9: Same comments about units and mentioning that FB method was used as for Fig. 7.

Figure 10: It would help to add the grazing period in either Fig 10 b) or c)

Figure 12: I suggest to replace "emission" with "flux" in y-axis for Figure a)

---

## Referee Comment (RC2) · Anonymous Referee #2 · 13 Dec 2018

General comments: The manuscript "Grazing related nitrous oxide emissions: from patch scale to field scale" uses scaled chamber N2O fluxes and eddy covariance N2O flux measurements to estimate field scale N2O emissions from dairy cow grazing systems in Switzerland. The authors used an established Lagrangian stochastic model to scale up chamber measurements from excreta and "background" fluxes. The scaling approach results are interesting and show the challenges of quantifying whole-farm N2O fluxes. The authors used gap filling approaches to fill gaps in their eddy covariance N2O flux dataset but there was not much discussion about the gap filling results. My suggestion is that this discussion should be expanded. In addition, it would be important to include more details in the methodology on the EC and chamber measurements and scaling approaches (see specific comments). Some sentences in the

text are difficult to understand, so the writing requires further work. I also noticed some grammar mistakes in the text. I recommend the authors to perform a thorough review of the manuscript to correct these mistakes before resubmitting the manuscript.

Specific comments

L9 – replace "the emissions" by N2O emissions

L10 – replace "variability" by "variabilities" and "and therefore emissions" by "so N2O emissions"

L11 – replace "of two grazing systems" by "for two grazing systems"

L12 – replacing "season 2016" by "season of 2016". In addition, I suggest including the number of dairy cows for each herd.

L15 – "Excreta patches and background surfaces on the pasture were identified manually". I suggest to be more specific here by saying that urine patches were identified based on the soil electric conductivity.

L20 – "(960 $\pm$ 219 g N2O-N, or 25 %)" This number is a little confusing. What does the 25% represent and shouldn't the emission units be expressed in per area?

L29 – replace "In the atmosphere, nitrous oxide" by "Nitrous oxide". In addition, include the appropriate citation for this sentence.

L30 – replace "it has a strong potential" by "N2O has a strong potential". I noticed that the replacement of nous by pronouns in some sentences throughout the text can compromise the clarity of those sentences. I suggest the authors to be as direct as they can in their sentences for the sake of clarity.

L31 – replace "to fertilization" by "to nitrogen fertilization"

L1 – "especially by cows". Are you referring specifically here to dairy cows? If so, please specify.

L2 – "excreta from the animals" replace by "from animal excreta"

L3 – replace "often even exceeds" by "often exceeds"

L3 to L5 – "Directly applied on a pasture soil…" this sentence is awkward and needs to be reworded.

L11 – "previously (Cardenas et al., 2010) and urine patches of cattle have exceptionally high loading" rates" by "previously (Cardenas et al., 2010). Urine patches of cattle have exceptionally high N loading rates"

L17 – replace "(e.g. EF of 0-14% of applied urine N, n=40; Selbie et al., 2015) and many of those studies measured the" by "(e.g. EF of 0-14% of applied urine N, n=40; Selbie et al., 2015). Many of those studies measured the". In addition, give some examples of the "many of those studies".

L18 – replace "conditions and thus these results are questionable with regard to" by "conditions making these results are questionable with regard to"

L20 – "these emissions" which emissions?

L20 - "(e.g. Arriaga et al., 2010)" provide more examples of studies and more the citation to the end of the sentence.

L21 – "For this purpose forage" replace by "For this purpose, forage"

L22 – "can be fed as a supplement to N rich grass and this subsequently leads to less" replace by "can be used as a feed supplement with N rich grass leading to less"

L24 – "real practice conditions". Do you mean real management conditions?

L23-24 – "experiments… are very rare". Cite some of the existing ones.

L26 – "a small scale" replace by "on a small spatial scale"

L26 –"and to attribute them to certain emission drivers" this statement needs to be reworded for clarity.

L30 – "by integration over a larger domain". Integration of what? Do you mean fluxes? Larger domain than chambers?

L30 – "It was already applied". What is "it"? The EC technique? If so, replace "it" by "The EC technique"

L32 – "(intensive – extensive, different crops, land / lake, . . .)" replace by "(intensive – extensive, different crops, land / lake, etc.)"

L33 – "for one system is preferable" replace by "for each system is preferable"

L2 – "to understand the contributions" replace by "to estimate the contributions"

L2 – "of the single emission sources" replace by "from single emission sources"

L4 – "emissions of two" replace by "emissions from two"

L5 – "diet for the cows" replace by "cows' diet"

L6 – The word fast should not be capitalized.

L8 to 9 – "We aimed at a better understanding of the quantity of the overall pasture emissions, the different emission sources and the reduction of corresponding uncertainties". This sentence is awkward and needs to be reworded.

L12 to 13 – provide the experimental period.

L13 – "detail in Voglmeier" replace by "detail by Voglmeier"

L14 – "annual average rain amount". Is snow also included in the total amount? If so, replace the word "rain" by "precipitation".

L15 to 16 – "(about 20 % clay, 35 % silt and 16 45 % sand" there is no need to show

this since this soil texture data are shown in Table 1.

L16 – "Soil measurements were performed...". Can you be more specific?

L18 – "the last renovation of the field in 2007" replace by "the last field renovation in 2007,"

L19 to 20 – "the fertilization rate was in the order of 120 kg N ha-1 20 per year between 2007 and 2015". Can you please specify the fertilization timing?

L23 – "12 cows per system.". Please reference figure 1.

L24 – "with additional maize silage". Was this silage offered to the cows in a different area? Did the silage supplementation influenced the time in which the cows spend in the grazing system?

L30 – "X indicating both systems". I suggest using M or G instead of X to avoid confusion.

L7 to 8 – "July and made a grass cut on the 22nd of June necessary" replace by "July and requiring a grass cut on June 22nd "

L15 – "For the comparison with the field-scale EC". Which comparison? Be more specific.

L25 – "to perform chamber measurements attributable to distinct surface conditions" replace by "to estimate N2O emissions from different surface sources".

L30 to 31 – "GS3 probe 31 (Meter Group, US" replace by "soil probe (GS3, Meter Group, US)".

L1 to 2– "Conductivity values exceeding a threshold of 0.15 mS cm-1 1 2 were marked as possible urine patches for further chamber measurements." It is important to explain

how this electric conductivity threshold was established.

L 3 – "Exemplary" replace by "Time series of"

L10 – "taken mainly during dry soil conditions" Can you provide the soil water content associated with "dry soil conditions"?

L18 – "The gas 18 was sucked through" replace by "The gas 18 was drawn through"

L18 – "a 40 m 1/4" PA tube allowing". Use metric units do express the dimensions of the tubing. Does 1/4" refer to the internal diameter of the tube? Please specify. What does "PA" stand for?

L19 – "The sample flow rate Q was typically around 8 l min-1 19". Did you use a mass flow controller to keep the flow rate constant?

L21 – "foam material to avoid uncontrolled air exchange". Was the chamber covered with some insulating material? What was the typical temperature differences within and outside the chamber during these measurements?

L21 to 22 – "The chamber was also equipped with a GMP343 (Vaisala, FL) $CO_2$ probe to measure the soil respiration." Do you show this $CO_2$ data? If not, I suggest excluding this sentence.

L22 to 23 – "The increase in concentration after placing the chamber on the soil was recorded every three seconds for a time period of about 90 seconds." For your chamber flux calculations, did you take into account the time necessary to purge this long tube right after the sampling line was connected to the analyzer?

L2 – "(slow chamber volume exchange and short measurement time)". Can you provide an average value for the chamber volume exchange?

L13 to 14 – "a thermocouple for air temperature measurement within the chamber, a

GS3 probe (see Sect. 2.4.1) and a ML3 Thetaprobe (Delta-T Devices Ltd, UK) for soil moisture and temperature observations (c. 0-5 cm and 0-10cm depth, respectively)." This sentence is a little confusing and needs to be reworded.

L4 – "were fenced to avoid unwanted animal contact". Can you provide the area of the fenced area around the tower?

L9 – Does this sonic anemometer infers the air temperature based on the sonic temperature or it has its own temperature sensor?

L10 – replace "sucked" by "drawn"

L11 – Please provide the pore size of the filters

L16- "The sample frequency of the EC system was generally 10 Hz". Does this mean that there was variation in the sample frequency? Why is that?

L18-19 – This sentence is awkward and needs to be reworded.

L22 – "The approach is based on. . ." What approach are you referring to?

L24 – 500 data points?

L24 – replace "double rotation" replace by "double coordinate rotation"

L28 – "several seconds". Provide the typical time lag value and its standard deviation.

L31 – "a time window of 0.61 seconds". How was the number determined

L1 – "In order to minimize the effect of non-stationarities in the time series, the 30 min flux was finally calculated as average over six 5 min subinterval flux values.". I wonder what would be the effect of this averaging approach on the low frequency spectral losses of their EC system. Furthermore, if you are already screening the data for non-

stationarity (page 8 L24) why to estimate fluxes for these short time intervals?

L7 – "half-hourly damping factors". Do you mean dampening factor?

L9 – "damping factors" see comment above

L10 – "damping effect" see previous comment

L11 – "EC flux measurements were taken" replace by "The EC flux was measured"

L19 – replace "which often result" by ", which often resulted"

L20 – delete the word "necessitate"

L21 – replace "like" by "for example"

L22 – replace "power break down" by "power outage"

L22 to 23 – replace "a data loss" by "data losses"

L24 – "(280o - 25o; 97o – 195o)" replace by (wind dir. = 280o - 25o and wind dir. = 97o – 195o)

L28 – "It was driven". What is "it" referring to?

L15 – "and it has to be checked". What is "it" referring to?

L18 – ", it is based on" replaced by ", based on"

L19 to 21 – Place the variable definitions after the equation.

L22 – "80' 000 trajectories were released backwards in time" replaced by "80,000 fluid particles were released backwards in time". Also, what is the time scale of this simulations? 30-min periods?

L24 – "systematic uncertainty". Do you mean "accuracy"?

L1 – I think section 2.6 is out of place. It should come after section 2.7.

L2 – what is the datalogger model used in this study?

L6 – In this section, it would be important to provide the spatial resolution of the grid used for upscaling the chamber fluxes. More details are also necessary on how the authors went from the output of Eq. 2 to the scaled fluxes. Did you generate digital maps of source emissions and then overlapped these maps with a footprint map? What was the software used to do these calculations?

L13 – "patches, . . .)" replace by "patches, etc.)"

L15 – "Upscaling FB measurements to the EC footprint was performed" replace by "FB measurements were up scaled to the EC footprint"

L16 – replace "contribution" by "contributions"

Page 11 L3 – "Upscaling to the paddocks" replace "Upscaling fluxes to the paddocks"

L10 – "Occasional negative individual flux values". What is the detection limit of this EC system? I think this would be an important variable to know to interpret these fluxes.

L3 - "Fluxes of background and dung patches were significantly smaller". Did you perform a statistical test to support this statement?

L 4 – "Especially fresh deposited urine patches with excreta ages below 3 days were able to emit more than" replace by "Freshly deposited urine patches under 3 days old could result in N2O emissions larger than"

L26 – "exponential decrease" replace by "exponential function"

L25 – "the variations were less pronounced". Which variations were less pronounces.

L 19 – "The good agreement btween the two independent approaches" provide a statistical index to support this statement.

L20 – replace "incl." by "including"

L28 to 29 – This sentence is a little confusing and needs to be reworded.

L21 – "significant sytem difference". Over which period of time and shouldn't this difference be expressed per area?

L3 to 4 – "e.g. N2O emissions related to the maize production...". Could you include values in the literature typical emission factors for corn silage production? These data would allow a fair comparison between the two grazing systems.

L9 – "They are based on". Specify who are "they".

L18 – "looking only at" replace by "looking only into"

L18 – "lead to increased soil" replace by "leads to increased soil"

L23 – "emission optimum". What does the word "optimum" mean here? Low N2O emissions?

L14 – replace "prohibited" by "prevented"

L2 – "N of the Swiss dairy" replace by "N provided by the Swiss dairy"

Table 5 – "EC integral system emission EC". Do you mean: Integral EC flux system emission?

[Figure]

---

## Author Comment (AC1) · 15 Jan 2019

**bg-2018-435**

**Author response to comments of referee #1**

We'd like to thank reviewer #1 for his careful and detailed review and appreciate his valuable comments.

In the following list, the referee comments are printed in *italic* and author responses are printed in blue. For the majority of the minor (language related) comments we fully adopt the referee suggestions. Below only the comments are listed that required a specific response.

**General comments**

1. However, one main point that needs to be addressed is the final conclusion; I strongly disagree that you can conclude that an optimised diet (with additional maize silage) in system M leads to a 25% reduction effect for N2O emissions based on the smaller area needed for grazing. The authors need to take the N2O emissions related to the maize production used in the diet into account; otherwise this comparison is not valid.

It needs to be noted here, that we only considered the  $N_2O$  emissions related to the cow excreta on pasture in this study. This is in line with the IPCC concept for emission factors and inventory calculations that generally relate the  $N_2O$  emissions to specific N inputs (see Introduction line 8-10). The comparison of different agricultural production systems with full accounting of the production chain (life cycle assessment) is beyond the scope of this study and will be published elsewhere.

Yet we agree that the use of the term "system" and the corresponding phrasing was not always clear and consistent. We will rephrase the conclusion to clarify that the statement "the emissions showed a clear difference of about 25 % between the two systems" specifically concerns the N2O emissions related to the cow excreta on the pasture. As mentioned in the conclusion, this demonstrates the mitigation potential of a N-reduced (N-optimized) feeding strategy. The latter does not necessarily require supplement maize silage feed but may also be achieved by an improved energy to protein ratio of the pasture grass.

2. Generally the authors need to be more careful with figure and table captions. The structure of some tables needs to be improved and the authors need to be more carful with units, especially when presenting cumulative fluxes (table 5). There are many abbreviations that were not explained (e.g. ECM, FAD, Q, A, V) or not very clearly (FD, FU, FU, temp, Fbg), which makes equations difficult to understand (section 3.2.2 and 3.2.3). Improving figures and tables and explaining abbreviations will help to make the manuscript easier to read.

We agree with the reviewer and will take his suggestions (see detailed comments below) for improving figures and tables into account. Furthermore, we will explain all abbreviations in more detail.

**Detailed comments**

P1, line 26/27: This conclusion is not correct, as the N2O produced during the production of Maize fed to the animal is not included in the calculations! See answer to general comment #1 (above).

**Introduction:**

*P1, line 30: Please add a reference for the GWP of N2O.* A reference will be added (IPCC, 2014).

**P2, line 6: Please insert "from excreta" after "N loading. . . was shown previously" and insert N after exceptionally high.."**

"From excreta" would not be correct here as the referenced study showed the effect of different N loading rates of inorganic fertilizer nitrogen on  $N_2O$  emissions.

The "N" was inserted as suggested.

**P2, line 16: Please give a suggestion of how emissions could be reduced if individual contributions are better understood.**

We will add the following sentence: "A better understanding of the individual contributions would also be very helpful to reduce the emissions, as e.g. dietary changes typically affect the excreted urine N which is mainly responsible for the high  $N_2O$  emission associated to excreta (Dijkstra et al., 2013)."

**Material and Methods:**

*P3, line 15: use average values for clay, silt and sand from table 1* We will follow the suggestion of referee #2 and only keep the reference to Table 1 (omitting the values in the main text).

P3, line 17: The range of 10-50 and 7-40 % of Lolium and Trifolium is quite large; could you give an average  $\pm$  stdev and the method of how it was assessed?

We will add the following information: "The vegetation consisted of a grass-clover mixture typical for Swiss pastures ( $78 \pm 12$  % grasses and  $15 \pm 10$  % legumes; main species: *Lolium perenne* and *Trifolium repens*, 10 sampling times between May and September).

P3, line 25: You write that the optimized protein content reduced the N input to the pasture. Did you measure this? Otherwise, just write that it was expected to reduce the N input to the pasture. We would like to keep the sentence unchanged. The reduction was calculated with the animal N budget model. That model (as explained in Sect. 2.3) uses measured data (e.g. milk yield, body weight gain, feed N contents) to calculate the excreted N.

**P4, line 18: Please add the name of the model and give a reference**

Unfortunately, there exists no official name or model reference. However, the model is based on the work by Bracher et al. (2011) and is used for the calculation of the Swiss greenhouse gas inventory. The model was used in other studies as well (Bell et al., 2017; Voglmeier et al., 2018). We will include the reference to Bracher et al. (2011).

**P4, line 20: Move this sentence up to the end of the sentence (line18-20)* After rephrasing the previous sentence, we would like to keep the sentence position unchanged.**

P5, line 10: change (Fig.2) to (Fig.2b)Will be changed accordingly (but to Fig. 2a → soil moisture).

P5, line 20-21: I am not quite sure if I understand this modification. Did you add a vent to the box? Then better to call it vent than inlet as the inlet is connected to the QCL. Please be more specific: I assume the 4 cm is the diameter and the 1m is the length of the vent tube and the 10 cm is the length of the foam material within the tube? What is the foam made of?

We agree with the reviewer that the modification was not properly described. As the referee assumed, a vent tube (to ambient air) was added instead of a fully closed circuit originally used in the Fast-Box. We will be more specific and describe the vent tube in more detail in the revised version.

*P5, line 22: As you don't show or discuss any soil respiration measurements, I suggest to delete this information.*

We would like to keep this information, as the  $CO_2$  soil respiration was used as a proxy to check if the chamber was properly sealed (as discussed in Sect. 2.4.3).

P5, line 24: Insert "the" after "The inflow off. . ." I assume by "the inlet" you mean " the vent"? As the FB chamber is a closed dynamic system (acc. to Hensen et al.2006). Will be changed accordingly (see also response above).

*P5 line 26: please add explanations to all abbreviations used in equation 1 (V, A, Fcham)* Will be added accordingly.

*P6 line 12: please give the make of the thermocouple* Will be added in the revised version.

P7 line 24: 500; add unit Will be added ("500 data points").

*P8 line 12: change (see Fig 2) to (see Fig. 2c). Please show the harvest event in the Figure (see comments to Figures)* Will be changed to Fig. 2c. We will show the harvest event in the Figure (as described in the answers

to the Figures).

*P8 line 33: Can you give a time period for the soil temperature classes?* The time period spans over the GOP. We will include this information.

P11 line 8/9: This last sentence is not clear. Please clarify which fluxes you are talking about (individual emission source; paddock or system M/G?) We will omit this sentence as it is not necessary.

**Results:**

P11 line 12: "they varied significantly"; significantly different from what? Background fluxes? The half-hourly fluxes showed generally a high variability during the grazing season. We will rephrase the sentence and state our intention more clearly.

P11 line 13: mention the harvest, were they increased after the harvest? There was a short increase in  $N_2O$  emissions directly after the harvest event. We will include this information in the text. P12 line 15-19: In Figure 8 FU,temp and FD,temp are fluxes averaged over 3 days, while in the text you are describing average daily values (?), which is confusion. I suggest to show average daily values in Figure 8, or use different abbreviations (e.g. FU,temp3d vs FU,temp1d), or only discuss 3d averages in the text. Its not clear what the "absolute highest FUtemp" is (5117 ug N2O-N m-2h-1), if the highest average value is 660 ug N2O-Nm-2h-1.

We agree with the referee that the paragraph is a bit confusing. We will rephrase it and only describe the 3-day averages plus individual single fluxes.

P12 line 20: you mention that Dung related emissions "showed a relation to excreta age", please mention what kind of relation. Please change "dung patch emissions" to "dung patch fluxes" The relation can be seen in Fig. 8. We will add a reference to the Figure. Additionally we will change "dung patch emissions" to "dung patch fluxes".

P12 line 25: Were the background fluxes not also sign. smaller compared to dung patch emissions? Looks like it in figure 8.

The background fluxes are also sign. smaller than dung patch emissions. We will rephrase the sentence to "...were significantly smaller compared to the excreta emissions".

**P12 line 32: How do you justify to set negative values to zero?**

The measured 3 days averages are all above zero. We only set the corresponding values of the fitted function (second order polynomial) after about 25 days to zero. The function as used in our study (Fig. 8, Eq. 4) would always result in negative values, which would not be representative for the measured values anymore.

P13 line 6: As "FD,temp" is not influenced by environmental conditions it equals FD (?)This should be stated here.

We will include this information.

P13 line 7-14: Please define the three sectors. I suggest to insert (<0.27, 27-33, >0.33) after ". . .by three different VWC sectors." It would help to show this in a graph.

We follow the referee's suggestion and insert (<0.27, 27-33, >0.33) after "... VEC sectors". However, we do not want to include a graph as the paper already has quite a high number of graphs and the additional information provided by a graph would be comparatively low.

P13 line 13: do you mean "similar values" or comparable to what? Can you add a stdev? We meant similar values. The values including the stdev would be 12 ± 3  $\mu$ g N2O-N m-2 h-1. We will include this information in the text.

*P14, line 17: It's not clear where the grazing period ends, therefore please add this information into the table (see comments table 10b).*

We assume, that the referee means Fig. 10b. We will include this information in an updated figure.

P14 line 25-27: these two sentences are not very clear. What do you mean by variations? The magnitude of fluxes varied less? I suggest to replace "rather limited" with "less pronounced" With variations, we meant the temporal variability of the magnitude of the fluxes (the blue dots in Fig. 10c versus the red dots in Fig. 10c). We will rephrase the sentence to "Nevertheless, the temporal

variability of the up-scaled 30 min FB fluxes was less pronounced". Furthermore, we will rephrase "rather limited" with "rather weak".

P15 line 2: But in Figure 11 it looks like fluxes were slightly higher for up-scaled FB fluxes. We agree with the referee on this mistake. We will correct it by replacing "...slightly lower..." by "slightly higher".

**Discussion:**

**P15, line 20/21: I don't understand this sentence**

We agree with the reviewer, that the sentence was confusing and thus we will rephrase it as follows: "We assume that the EC fluxes are on average representative for the whole pasture system, although the contribution of the central paddocks X.11, X.12 and X.21, X.22 to the EC footprint is generally higher than the contribution of the other more distant paddocks (Fig. 4).

P15, line 25: Include "(data not shown)" at the end of "other characteristics" as you didn't show any productivity (yield). Delete "Also" at the beginning of the next sentence.Will be changed accordingly.

P15, line 28-p16 line 4: This paragraph is difficult to understand. I strongly disagree that you can conclude that an optimised diet in system M leads to a 25% reduction effect for N2O emissions based on the smaller area needed for grazing. You need to take the N2O emissions related to the maize production into account, otherwise this comparison is not valid. See answer to general comment #1.

P16, line 23: 1.03 kg N2O-Nha-1y-1, please explain in more detail how this value was calculated. This value should have been shown in the results section 3.3.2.

The value was calculated by using Eq. 5 (as stated) similar to the cumulative background emission in Fig. 12b (green area) but for the entire year using measured soil moisture. We will explain this more clearly in the text. This full year extrapolation value was only calculated for comparison with literature values. Therefore we think, it should not be part of the result section, as this section focuses on the grazing period (between March to November).

**P16, line 26/27: This last sentence is out of context**

We intended to provide a second way how these background fluxes on pastures can be stimulated. Nevertheless, we agree with the referee that this was not clearly stated and thus we will rephrase the sentence.

"On pastures, these background emissions can also result from trampling of the cows which can further stimulate the N2O production via denitrification due to soil compaction (Bhandral et al., 2007)."

Tables:

Table 1: As soil depth is not really a parameter I suggest to re-arrange the table; one column for each soil depth, with missing values in each column as the different parameters have not been measured in all soil layers

We will try to re-arrange the table for better readability.

*Table 2: What does ECM stand for? Please explain abbreviation (maybe in footnote).* It is the energy corrected milk. We will add this abbreviation in the table caption.

Have the animals been weighted before and after the experiment? Was the weight increase considered in the calculations of the excretion (heavier animals will excrete more)?

The animals have been weighted on a daily basis and the weight increase is considered in the calculations of the excretions. This is roughly described in Sect. 2.3 and in full detail in the referenced article by Voglmeier et al. (2018) for the same experiment. We will include more information on the animal measurements in Sect. 2.3.

Table 3: Please add information of flux measurement method (FB) Will be added accordingly.

Table 4: Please describe what the different equations are: Parameterisations of 3 day average fluxesfrom EB measurements, split into background, dung and urine fluxes.

We will add this information.

"The equations are based on 3-day average fluxes from FB measurements on different emission sources (background, urine and dung patches)."

Table 5: I assume that you are showing cumulative fluxes. You need to mention this together with the time scale (per GOP?). I suggest to simplify the table by only having two parts; add a dotted line above the N input and to move the FB urine and FB dung fluxes above the EFs. Please add N input from dung. What does FAD stand for?

We will add the information about cumulative fluxes and the time scale. Furthermore, we will try to simplify the table as suggested. We can add the N input from dung, but actually wanted to keep that out of the table as it can simply be calculated from N total – N urine. FAD was a term used in a development version of the manuscript and thus will be deleted.

Table 5: What is "EC integral system emission EC," ? Reading in the text (P15, line 28-31) I have the impression these fluxes are up-scaled FB fluxes to the whole system. If they are EC emissions, please describe more carefully in the text. The unit is confusing as it is an emission (concentration per area per time). Reading in the discussion I understand what you mean, but in the table it's not clear. Maybe you can explain in a foot note. "EF total" is calculated from EC, while "EF urine" and "EF dung" are calculated from up-scaled FB measurements (or not?) This needs to be stated clearly.

The value/units represent integral emissions for the entire pasture area and the investigated grazing period. In order to prevent confusion and misunderstandings, we will change the units of this type of results (also in the abstract and in the main text) and express them in units of  $N_2O$  -N cow-1 h-1, which has an equivalent meaning.

In addition we will reorganize the Table in a more logical way and include a footnote with detailed information on the listed quantities.

Figures: Figure 1: P29 line4 Insert "(triangles)" after "the two EC towers. . ." Will be added accordingly.

Figure 2 b): Move the legend, or change the scale so the bars for the high precipitation events in June and July are not cut off. Will be changed accordingly.

*Figure 2d): Add arrows for fertiliser application dates and for harvest date* We will add this information on Fig. 2c.

*Figure 3: Change the area showing a) to being transparent* Will be changed accordingly.

Figure 5: Please explain the reason for dotted frame. It's confusing that the lines connecting #dung and # urine patches to "Paddock flux dung patches" and "paddock flux urine patches" cross the arrows leading to "paddock flux background" and "paddock flux urine patches", it looks like they are feeding into them as well. Try to show clearer (maybe with a curved line over the crossing line). The dotted frame indicates a further processing step. We see, that this is actually not needed and will instead use the standard frame. We agree with the reviewer, that the crosses are a bit confusing and thus we will update the graph (as suggested, probably with a curved line).

Figure 6: Add arrows for exact fertilisation and harvest events. Add the date of the skipped value. It would be good to include the information of grazing periods in the graph to explain the increased fluxes. We will add arrows for the exact fertilization and harvest events and we will add the date of the skipped value. Furthermore, we will add the information that for the analysis of grazing related emission, the non-shaded periods of Fig.6 were used. Detailed information of the rotational grazing regime is already displayed in Fig. 3c.

Figure 7: I suggest to change the unit to ug N2O-N m-2h-1, it makes it easier to read the values in the graph and to compare to values you describe in the text (for Figure8, where ug N2O-N m-1h-1 are used). In the legend insert "from different sources" after "the comparison of fluxes.." and add the information that the fluxes were measured with FB technique.

We agree with the referee and we will change the units and add the information on the flux measurement technique in the legend.

Figure 8: Same comments about units as for Fig. 7. Please add in legend that fluxes were measured with FB. Are there any standard errors for the background fluxes, or were they too small to be seen? I suggest to change to x-axis description to "age of excreta [d]". Give information about the fitted curves (refer to equation 3+4).

We will change the units and the x-axis description to "age of excreta [d]" and we will add a reference to Eq. 3, 4.

There are standard errors for the background fluxes, but these are too small to be seen. We will add this information in the legend.

*Figure 9: Same comments about units and mentioning that FB method was used as for Fig. 7.* We will change the units as requested.

*Figure 10: It would help to add the grazing period in either Fig 10 b) or c)* The grazing phase spans over the full time scale as indicated in the legend. The up-scaled paddock fluxes of four single paddocks can be seen in Fig. 10b.

Bell, M., Flechard, C., Fauvel, Y., Häni, C., Sintermann, J., Jocher, M., Menzi, H., Hensen, A. and Neftel, A.: Ammonia emissions from a grazed field estimated by miniDOAS measurements and inverse dispersion modelling, Atmospheric Meas. Tech., 10(5), 1875–1892, doi:10.5194/amt-10-1875-2017, 2017.

Bhandral, R., Saggar, S., Bolan, N. and Hedley, M.: Transformation of nitrogen and nitrous oxide emission from grassland soils as affected by compaction, Soil Tillage Res., 94(2), 482–492, doi:10.1016/j.still.2006.10.006, 2007.

IPCC, 2014: Climate Change 2014: Synthesis Report . Contribution of Working Groups I, II and III to the Fifth Assessment Report of the Intergovernmental Panel on Climate Change [Core Writing Team, R.K. Pachauri and L.A. Meyer (eds.)], IPCC, Geneva, Switzerland., 2014.

Voglmeier, K., Jocher, M., Häni, C. and Ammann, C.: Ammonia emission measurements of an intensively grazed pasture, Biogeosciences, (15), 4593–4608, doi:doi.org/10.5194/bg-15-4593-2018, 2018.

---

## Author Comment (AC2) · 15 Jan 2019

**bg-2018-435 Author response to comments of referee #2**

We'd like to thank reviewer #2 for his careful review and appreciate his valuable comments, which significantly improved the manuscript.

In the following list, the referee comments are printed in *italic* and author responses are printed in blue. For the majority of the minor (language related) comments we fully adopt the referee suggestions. Below only the comments are listed that required a specific response.

**General comments**

The authors used gap filling approaches to fill gaps in their eddy covariance N2O flux dataset but here was not much discussion about the gap filling results. My suggestion is that this discussion should be expanded.

We used only one approach (LUT) for the gap filling of our EC fluxes, as mentioned in Section 2.5.3. This approach was chosen based on the cited evaluation by Mishurov and Kiely (2011) who discussed different gap filling approaches is more detail. It was not the objective of this study to assess the performance of gap filling approaches. However, we used the variability between LUT and three other approaches for estimating the uncertainty of the gap filling procedure.

In addition, it would be important to include more details in the methodology on the EC and chamber measurements and scaling approaches (see specific comments). Some sentences in the text are difficult to understand, so the writing requires further work. I also noticed some grammar mistakes in the text. I recommend the authors to perform a thorough review of the manuscript to correct these mistakes before resubmitting the manuscript.

We think, the methodology on the EC and chamber measurements (including scaling approaches) is already described very extensively with about 6.5 pages (excluding figures). Nevertheless, we intend to include most of the specific comments on methodology issues (see answers to specific comments below).

Regarding the language related issues, we agree with the reviewer and will perform a thorough review to correct these mistakes.

**Specific comments**

**Page 1**

L12 – replacing "season 2016" by "season of 2016". In addition, I suggest including the number of dairy cows for each herd.

Will be changed accordingly. We will include the number of dairy cows (12 for each herd).

L15 – "Excreta patches and background surfaces on the pasture were identified manually". I suggest to be more specific here by saying that urine patches were identified based on the soil electric conductivity.

We will rephrase the sentence to "After different grazing rotations, background and urine patches were identified based on soil electric conductivity measurements while fresh dung patches were identified visually. The magnitude and temporal pattern of these single emission sources were measured with a Fast-box (FB) chamber".

 $L20 - "(960 \pm 219 \text{ g N2O-N}, \text{ or } 25 \%)"$  This number is a little confusing. What does the 25% represent and shouldn't the emission units be expressed in per area?

We agree with the reviewer that the number is a bit confusing. The value/units represent integral emissions for the entire pasture area and the investigated grazing period. In order to prevent confusion and misunderstandings, we will change the units of this type of results (in the abstract and in the main text) and express them in units of N2O -N cow-1 h-1, which has an equivalent meaning. The "25%" represent the relative change of emissions between the two grazing systems. This will be clarified by including the emission values of the two systems and rephrasing in the following way: "Neglecting emission periods influenced by fertilizer applications resulted in significantly higher grazing related emissions per cow and grazing hour in system G (0.270 g N2O-N cow-1 h-1) compared to system M (0.201 gN2O-N cow-1 h-1)."

L29 – replace "In the atmosphere, nitrous oxide" by "Nitrous oxide". In addition, include the appropriate citation for this sentence.Will be changed accordingly and we will add a reference to the IPCC (2014).

L30 – replace "it has a strong potential" by "N2O has a strong potential". I noticed that the replacement of nous by pronouns in some sentences throughout the text can compromise the clarity of those sentences. I suggest the authors to be as direct as they can in their sentences for the sake of clarity.

Will be changed accordingly. Furthermore, we will try to locate those replacements and use the proper nouns.

**Page 2**

L1 – "especially by cows". Are you referring specifically here to dairy cows? If so, please specify. No, we refer to cows in general.

L3 to L5 – "Directly applied on a pasture soil. . ." this sentence is awkward and needs to be reworded. We will reword the sentence to:

Once applied to the pasture soil, the reactive nitrogen of excreta is transformed by microbial nitrification and denitrification processes and significant amounts of N2O can be produced as a by-product."

L17 – replace "(e.g. EF of 0-14% of applied urine N, n=40; Selbie et al., 2015) and many of those studies measured the" by "(e.g. EF of 0-14% of applied urine N, n=40; Selbie et al., 2015). Many of those studies measured the". In addition, give some examples of the "many of those studies". Will be changed accordingly and we will give some examples of those studies (Bell et al., 2015; Chadwick et al., 2018).

**L20 – "these emissions" which emissions?**

We will rephrase the sentence to make it clear that we mean the emissions associated to animal excreta. "The more efficient use of fed N is essential to reduce the emissions associated to animal excreta. "

**L20 - "(e.g. Arriaga et al., 2010)" provide more examples of studies and more the citation to the end of the sentence.**

We will provide more examples and put the citations at the end of the sentence. "...N excreted by the animals (e.g. Arriaga et al., 2010; Dijkstra et al., 2013; Yan et al., 2006)."

**L24 – "real practice conditions". Do you mean real management conditions?**

We will rephrase the sentence as follows: "...but corresponding emission experiments under real grazing conditions for a full season, to our knowledge, have not been reported hitherto."

L23-24 – "experiments. . . are very rare". Cite some of the existing ones. Actually, to our knowledge, no comparable experiment exists. We will rephrase the sentence. See also previous comment.

L26 – "and to attribute them to certain emission drivers" this statement needs to be reworded for clarity.

We will rephrase to "...to attribute the measured fluxes to potential emission drivers...".

L30 – "by integration over a larger domain". Integration of what? Do you mean fluxes? Larger domain than chambers?

We will rephrase to "...by integrating fluxes over a larger spatial domain."

L8 to 9 - "We aimed at a better understanding of the quantity of the overall pasture emissions, the different emission sources and the reduction of corresponding uncertainties". This sentence is awkward and needs to be reworded.We will omit this sentence, as it is redundant.

L12 to 13 – provide the experimental period.

We will provide this information (grazing period 2016).

L14 – "annual average rain amount". Is snow also included in the total amount? If so, replace the word "rain" by "precipitation". Snow is also included, thus we will change "rain amount" to "precipitation.

L15 to 16 – "(about 20 % clay, 35 % silt and 45 % sand" there is no need to show this since this soil texture data are shown in Table 1. Will be changed accordingly.

L16 – "Soil measurements were performed. . .". Can you be more specific? This sentence is referring to the preceding sentence (with reference to Table 1). We will rephrase the sentence to: "The sampling for analysis of soil texture and other soil characteristics of different layers were performed at four locations on the pasture in 2013 and 2016."

L19 to 20 – "the fertilization rate was in the order of 120 kg N ha-1 per year between 2007 and 2015". Can you please specify the fertilization timing?

We think, listing the timings of all previous fertilization events since 2007 is unnecessary in the context of the goal of the manuscript (grazing related  $N_2O$  emissions). The timing of the fertilisation events of the study year is included in Fig. 2d and will also be added to Fig. 6 in the revised version.

L23 – "12 cows per system.". Please reference figure 1. We will include a reference to Fig. 1a.

L24 – "with additional maize silage". Was this silage offered to the cows in a different area? Did the silage supplementation influenced the time in which the cows spend in the grazing system? The silage was fed in the barn when the cows had to go there for milking twice a day. In order to avoid an influence of the supplement feeding on the grazing time, the barn and grazing times were always fully synchronous for both herds/systems.

L30 – "X indicating both systems". I suggest using M or G instead of X to avoid confusion. We would like to keep the X because it simplifies the text considerably when referring to both systems equally.

**Page 4**

L15 – "For the comparison with the field-scale EC". Which comparison? Be more specific. We agree with the reviewer that the sentence is not specific enough. We will rephrase the sentence to "Moreover, the comparison between the field-scale EC method and the small scale chamber measurements required an estimate of the number of dung and urine patches on the pasture."

**Page 5**

L1 to 2– "Conductivity values exceeding a threshold of 0.15 mS cm-1 were marked as possible urine patches for further chamber measurements." It is important to explain how this electric conductivity threshold was established.

We will add some more information in the text. The threshold of 0.15 mS cm-1 was chosen based on pre-experimental tests with artificially applied urine patches on the pasture and areas not affected by grazing for a few month (background). The value of 0.15 mS cm-1 was determined as the maximum of the observed background conductivity, but was still far below the observed conductivity of fresh urine patches (see also Fig. 3).

L10 – "taken mainly during dry soil conditions" Can you provide the soil water content associated with "dry soil conditions"?

There is no fixed water content threshold for dry soil condition. Nevertheless, as can be seen in Fig. 2 and Fig. 9, we refer to volumetric soil moisture contents below roughly 0.4, thus clearly lower compared to the ones before July. We will add this information in the revised manuscript.

L18 – "a 40 m 1/4" PA tube allowing". Use metric units do express the dimensions of the tubing. Does 1/4" refer to the internal diameter of the tube? Please specify. What does "PA" stand for? We will add information as follows: "The sample air was drawn continuously from the FB headspace through a 40 m long polyamide tube (perfluoroalcoxy, O.D. 1/4") to the analyser …" We would like to keep the tube diameter in inches, as this is the official commercial labelling of this product.

**L19 – "The sample flow rate Q was typically around 8 l min-1". Did you use a mass flow controller to keep the flow rate constant?**

No, we did not use a flow controller. The flow rate was controlled by the controlled pressure (30 Torr) in the QCL analyser cell and a flow restrictor needle valve at the QCL inlet. The inlet tube represented an additional flow resistance. Since the effect of the valve and the tube were constant over time, the flow also remained quite stable.

L21 – "foam material to avoid uncontrolled air exchange". Was the chamber covered with some insulating material? What was the typical temperature differences within and outside the chamber during these measurements?

No, the chamber was not covered with some insulating material. Nevertheless, as the measurements with a fast-box are very quick (typically within 1-2 minutes), the temperature differences between within and outside the chamber stayed typically below 0.5 °C. Starting a new measurement, the chamber volume was always flushed (by tilting the box by 90 °) until the chamber volume was completely mixed with the air outside the box.

L21 to 22 – "The chamber was also equipped with a GMP343 (Vaisala, FL) CO2 probe to measure the soil respiration." Do you show this CO2 data? If not, I suggest excluding this sentence. We would like to keep this information, as the CO2 soil respiration (from the CO2 concentration increase) was used as a proxy to check if the chamber was properly sealed (as discussed in Sect. 2.4.3). L22 to 23 – "The increase in concentration after placing the chamber on the soil was recorded every three seconds for a time period of about 90 seconds." For your chamber flux calculations, did you take into account the time necessary to purge this long tube right after the sampling line was connected to the analyzer?

Yes, this time was taken into account. In addition we discarded the first and the last 5 seconds of the closure period in the time series from the regression analysis.

**Page 6**

L2 – "(slow chamber volume exchange and short measurement time)". Can you provide an average value for the chamber volume exchange?

The average volume exchange time is about 40 min. We will add this information.

L13 to 14 – "a thermocouple for air temperature measurement within the chamber, a GS3 probe (see Sect. 2.4.1) and a ML3 Thetaprobe (Delta-T Devices Ltd, UK) for soil moisture and temperature observations (c. 0-5 cm and 0-10cm depth, respectively)." This sentence is a little confusing and needs to be reworded.

We will reword the sentence.

**Page 7**

*L4* – "were fenced to avoid unwanted animal contact". Can you provide the area of the fenced area around the tower?

The fence was in a distance of about 2m around the tower in the main wind direction sectors. In the direction where the QCLs were stationed (trailer at system M, shelter at system G), a larger area was fenced (see white area around EC tower positions in Fig. 1a).

L9 – Does this sonic anemometer infers the air temperature based on the sonic temperature or it has its own temperature sensor?

It infers the air temperature based on the sonic temperature.

**L11 – Please provide the pore size of the filters**

The Midisart 2000 has a filter pore size of 0.2  $\mu m$  and the AcroPak has a filter pore size of 0.2  $\mu m$ . We will include this information in the text.

**L16- "The sample frequency of the EC system was generally 10 Hz". Does this mean that there was variation in the sample frequency? Why is that?**

We agree with the reviewer about the confusing sentence. The EC system was always operated at 10 Hz.

**L18-19 – This sentence is awkward and needs to be reworded.**

We will reword the sentence to "Additionally the program visualized the measurements of the  $N_2O$  concentrations and fluxes, calculated with an preliminary online flux calculation. The program also allowed to check the EC system by remote access.".

L22 – "The approach is based on. . ." What approach are you referring to?We refer to the customized program mentioned in the previous sentence. We will rephrase the sentence.

L24 – 500 data points? Yes, we meant 500 data points.

L28 – "several seconds". Provide the typical time lag value and its standard deviation. The typical time lag was about 6 seconds for system M and about 7 seconds for system G. We will add this information in the manuscript. But it is not possible to give a meaningful standard deviation of the 'dynamic' lags determined by the peak position in the cross-covariance function. In many cases, the signal-to-noise ratio of the fluxes were small due to low emissions or non-stationarity. In this cases the 'dynamic' lags were often not meaningful and very large. Due to this reason we applied a lag window filter (see Page 7, line 31) and used an average default lag otherwise. Thus the selected good quality lags were by definition within a window of  $\pm 0.61$  s).

L31 – "a time window of 0.61 seconds". How was the number determined

The number was determined based on the variability of the determined dynamic lag times. We will include a description how the number was determined in the revised version of the manuscript.

**Page 8**

L1 – "In order to minimize the effect of non-stationarities in the time series, the 30 min flux was finally calculated as average over six 5 min subinterval flux values.". I wonder what would be the effect of this averaging approach on the low frequency spectral losses of their EC system. Furthermore, if you are already screening the data for non-stationarity (page 8 L24) why to estimate fluxes for these short time intervals?

The low frequency losses were in the range of 1-5 %, based on theoretical calculations (Kaimal cospectra and transfer function for block averaging) and on the comparison of 30 min and 5 min subinterval fluxes. The theoretical approach was used to correct the fluxes for this low-frequency damping effect.

Since a large part of the 30 min  $N_2O$  fluxes was more or less affected by non-stationarity effects, the use of the 5 min sub-interval fluxes (to obtain 30 min average fluxes) generally lowered the non-stationarity effects leading to a higher quality of the fluxes (see page 8, line 1-2).

**L7 – "half-hourly damping factors". Do you mean dampening factor?**

*L9 – "damping factors" see comment above*

L10 – "damping effect" see previous comment

We think, both terms can be used, but the term "damping" is more commonly used in the EC literature. Therefore we want to keep it here.

**L19 – replace "which often result" by ", which often resulted"**

We would prefer to use "which often result" to indicate, that this not only happened in the past but is an ongoing issue with EC measurements.

**L28 – "It was driven". What is "it" referring to?**

The sentence will be rephrased to "The occurrence of data gaps showed a diurnal pattern with stronger data loss during the night, which was driven by the wind pattern with typically stronger wind speeds during daytime and calm nights." in the revised manuscript.

**Page 9**

L15 – "and it has to be checked". What is "it" referring to? We are referring to the spatial dimension of the footprint. This will be changed in the revised version.

L22 – "80' 000 trajectories were released backwards in time" replaced by "80,000 fluid particles were released backwards in time". Also, what is the time scale of this simulations? 30-min periods? Yes, the footprint simulation time scale was 30 min. We will add this information in the text and replace trajectories by fluid particles.

**L24 – "systematic uncertainty". Do you mean "accuracy"?**

No, we mean the "systematic uncertainty", as described in the referenced articles at the end of the sentence.

L1 – I think section 2.6 is out of place. It should come after section 2.7. We think, it is actually not out of place as Sect. 2.7 builds on data retrieved from Sect. 2.6 (soil moisture / soil temperature measurements). Thus, we would like to keep the structure of the sections.

L2 – what is the datalogger model used in this study?

At system M (north), a Campbell Scientific CR10X data logger was used. At the system G (south), we used the Campbell Scientific CR1000.

L6 – In this section, it would be important to provide the spatial resolution of the grid used for upscaling the chamber fluxes. More details are also necessary on how the authors went from the output of Eq. 2 to the scaled fluxes. Did you generate digital maps of source emissions and then overlapped these maps with a footprint map? What was the software used to do these calculations? We are not completely sure, whether we understand the question. We did not use a grid to upscale the chamber fluxes to the EC system. Equation 2 was evaluated for each paddock (integrating the particle touchdowns within the respective paddock area) for each 30 min interval. This resulted in a footprint contribution for each paddock which was multiplied by the paddock scale emissions for urine dung and background as described in Fig. 5 and Section 2.7.

We will accordingly improve the description of the upscaling procedure in Sections 2.5.4 and 2.7.

**Page 11**

L10 – "Occasional negative individual flux values". What is the detection limit of this EC system? I think this would be an important variable to know to interpret these fluxes.

The negative fluxes exclusively resulted in cases, when no peak in the cross-covariance function could be identified (and thus the value at the default lag was used). Thus is can be concluded that the negative fluxes were generally below the detection limit (which was time dependent e.g. due to the varying influence of non-stationarity effects). We will add this information in the text.

**Page 12**

L3 - "Fluxes of background and dung patches were significantly smaller". Did you perform a statistical test to support this statement?

Yes, we performed the Student's t-test which resulted in p values < 1e-12. But we also think that the results plotted in Fig. 7 are clear in this respect.

**Page 14**

L25 – "the variations were less pronounced". Which variations were less pronounces. We will rephrase to "...the variability of the up-scaled FB fluxes were less pronounced.".

**Page 15**

L 19 – "The good agreement between the two independent approaches" provide a statistical index to support this statement.

We will expand the sentence to: "The good agreement between the two approaches (< 2% difference) ...". This difference is much lower than the uncertainty ranges also listed in Table 5.

*L28 to 29 – This sentence is a little confusing and needs to be reworded.* We will simplify the sentence for better readability.

L21 – "significant system difference". Over which period of time and shouldn't this difference be expressed per area?

We agree with the reviewer and will replace this value by the emissions per cow and grazing hour of system M and G (see comment to Page 1, line 20 above).

**Page 16**

L3 to 4 - "e.g. N2O emissions related to the maize production. . .". Could you include values in the literature typical emission factors for corn silage production? These data would allow a fair comparison between the two grazing systems.

It needs to be noted here, that we only considered the  $N_2O$  emissions related to the cow excreta on pasture in this study. This is in line with the IPCC concept for emission factors and inventory calculations that generally relate the  $N_2O$  emissions to specific N inputs (see Introduction line 8-10). The comparison of different agricultural production systems with full accounting of the production chain (life cycle assessment) is beyond the scope of this study and will be published elsewhere.

As mentioned in the conclusion, the emission difference between the two pasture systems demonstrates the mitigation potential of a N-reduced (N-optimized) feeding strategy. The latter does not necessarily require supplement maize silage feed but may also be achieved by an improved energy to protein ratio of the pasture grass.

**L9 – "They are based on". Specify who are "they".**

We meant the EFs. We will change that in the revised version of the manuscript.

**Page 17**

L23 – "emission optimum". What does the word "optimum" mean here? Low N2O emissions? No, we meant emission maximum. The term "optimum" is often used in this context (e.g. Butterbach-Bahl et al., 2013).

**Page 28**

Table 5 – "EC integral system emission EC". Do you mean: Integral EC flux system emission? The entry represents integral emissions for the entire pasture area and the investigated grazing period. In order to prevent confusion and misunderstandings, we will change the units of this type of results (also in the abstract and in the main text) and express them in units of N2O -N cow-1 h-1, which has an equivalent meaning.

Bell, M. J., Rees, R. M., Cloy, J. M., Topp, C. F. E., Bagnall, A. and Chadwick, D. R.: Nitrous oxide emissions from cattle excreta applied to a Scottish grassland: Effects of soil and climatic conditions and a nitrification inhibitor, Sci. Total Environ., 508, 343–353, doi:10.1016/j.scitotenv.2014.12.008, 2015.

Butterbach-Bahl, K., Baggs, E. M., Dannenmann, M., Kiese, R. and Zechmeister-Boltenstern, S.: Nitrous oxide emissions from soils: how well do we understand the processes and their controls?, Philos. Trans. R. Soc. B Biol. Sci., 368(1621), 20130122–20130122, doi:10.1098/rstb.2013.0122, 2013.

Chadwick, D. R., Cardenas, L. M., Dhanoa, M. S., Donovan, N., Misselbrook, T., Williams, J. R., Thorman, R. E., McGeough, K. L., Watson, C. J., Bell, M., Anthony, S. G. and Rees, R. M.: The contribution of cattle urine and dung to nitrous oxide emissions: Quantification of country specific emission factors and implications for national inventories, Sci. Total Environ., 635, 607–617, doi:10.1016/j.scitotenv.2018.04.152, 2018.

Mishurov, M. and Kiely, G.: Gap-filling techniques for the annual sums of nitrous oxide fluxes, Agric. For. Meteorol., 151(12), 1763–1767, doi:10.1016/j.agrformet.2011.07.014, 2011.

---

## Author Response (AR1)

**bg-2018-435**

**Author responses to comments**

We thank the two referees and the editor for their very careful review of the manuscript. Their detailed and valuable comments helped to significantly improve the manuscript. For the majority of the language related comments we directly adopted the referee suggestions. Below, the scientific comments are listed that required a specific response.

The referee comments are printed in *italic*, and the author responses are printed in blue. The editor comments are printed in *green* and have been attached to the respective referee comment they are referring to.

**Referee #1**

**General comments**

1. However, one main point that needs to be addressed is the final conclusion; I strongly disagree that you can conclude that an optimised diet (with additional maize silage) in system M leads to a 25% reduction effect for N2O emissions based on the smaller area needed for grazing. The authors need to take the N2O emissions related to the maize production used in the diet into account; otherwise this comparison is not valid.

EDITOR: I agree that one can easily think of the evaluation of the upstream processes or the full production chain. Therefore, it must be made very clear what the (limited) scope of this manuscript is, but in the discussion, it the restricted scope should be mentioned again and briefly discussed what else would be needed to evaluate the whole chain. This is necessary for the readers not to draw wrong conclusions. The conclusions should also refer very clearly to the limited scope of the study to avoid any misinterpretation.

It needs to be noted, that we only considered the  $N_2O$  emissions related to the cow excreta on pasture in this study (as indicated by the paper title). This is in line with the IPCC concept for emission factors and inventory calculations that generally relate the  $N_2O$  emissions to specific N inputs (see Introduction). As suggested by the editor, we additionally clarified the scope of the study in the different parts of the manuscript. Yet we agree that the use of the term "integral system emission" (in units of kg  $N_2O$ -N) in the original manuscript version was misleading. In order to prevent confusion, we thus changed the units of this type of results (used for comparison of the two grazing systems) to emissions per cow and grazing hour ( $N_2O$ -N cow-1 h-1), which have an equivalent meaning. Moreover, the direct comparison of the grazing related emissions of the two systems has been revised and is discussed in more details.

With the comparison of system M and G we mainly wanted to test whether an N-reduced feeding leads to reduced N excretion and N2O emissions on the pasture. In the conclusions, we have added the consideration, that an N-optimized feeding strategy does not necessarily require supplement maize silage feed but may also be achieved by an improved energy to protein ratio of the pasture grass. Moreover, it needs to be noted that the comparison of different agricultural production systems with full accounting of the production chain (life cycle assessment) is a complex concept (usually not directly related to field measurements) using many different assumptions and data sources, and is well beyond the declared scope of this study.

2. Generally the authors need to be more careful with figure and table captions. The structure of some tables needs to be improved and the authors need to be more careful with units, especially when presenting cumulative fluxes (table 5). There are many abbreviations that were not explained (e.g. ECM, FAD, Q, A, V) or not very clearly (FD, FU, FU,temp, Fbg), which makes equations difficult to understand (section 3.2.2 and 3.2.3). Improving figures and tables and explaining abbreviations will help to make the manuscript easier to read.

We agree with the referee and improved the manuscript in the mentioned respects as far as possible (see responses to detailed comments below).

**Detailed comments**

P1, line 26/27: This conclusion is not correct, as the N2O produced during the production of Maize fed to the animal is not included in the calculations! See answer to general comment 1 (above).

Introduction:

*P1, line 30: Please add a reference for the GWP of N2O.* A reference was added (IPCC, 2014).

*P2, line 6: Please insert "from excreta" after "N loading. . . was shown previously" and insert N after exceptionally high.."*

"From excreta" would not be correct here as the referenced study showed the effect of different N loading rates of inorganic fertilizer nitrogen on  $N_2O$  emissions. The "N" was inserted as suggested.

*P2, line 16: Please give a suggestion of how emissions could be reduced if individual contributions are better understood.*

We added the following sentence: "A better understanding of the individual contributions would also be very helpful to reduce the emissions, as e.g. dietary changes typically affect the excreted urine N which is mainly responsible for the high N2O emission associated to excreta (Dijkstra et al., 2013)."

**Material and Methods:**

P3, line 15: use average values for clay, silt and sand from table 1We omitted the redundant values in the text and only kept the reference to the values in Table 1.

P3, line 17: The range of 10-50 and 7-40 % of Lolium and Trifolium is quite large; could you give an average  $\pm$  stdev and the method of how it was assessed?

We added the following information: "The vegetation consisted of a grass-clover mixture typical for Swiss pastures ( $78 \pm 12$  % grasses and  $15 \pm 10$  % legumes; main species: *Lolium perenne* and *Trifolium repens*, 10 sampling times between May and September)."

P3, line 25: You write that the optimized protein content reduced the N input to the pasture. Did you measure this? Otherwise, just write that it was expected to reduce the N input to the pasture. Following the referee's suggestion we rephrased to: "This was supposed to reduce the excreta N input to the pasture." The excreta N input to the pasture could not be measured directly but it was inferred from the data based animal N budget (Sect. 2.3).

**P4, line 18: Please add the name of the model and give a reference**

EDITOR: In your response you imply that a satisfactory description of the model may not exist. Am I right? The argument that the model has been used in other published work does not substitute the apparent lack of the model description. Using a model in science requires the description of the theory and how it is implemented in the model code. Furthermore you need to refer to how the model was tested and whether or not it was supported by the tests. If you are not able to refer to a published description, you need to describe the model yourself or use another model. Consider including the model description in the appendix .- basic equations, drivers, parameters, outputs, consideration of uncertainty, tests.

The online response to this comment was unfortunately not adequate. The term 'budget model' used in the original version was obviously misleading and not really appropriate for the applied cow nitrogen budget. We rephrased Section 2.3 to clarify the used approach. In short, the N budget of a dairy cow simply balances the N input by feeds with the N-accumulation in the cow body (weight gain) and the N losses by milk yield and excreta output. Thus the excreted N amount can be inferred from the other 3 terms. This calculation and the corresponding uncertainty has already been presented in the previous paper by Voglmeier et al. (2018) for the same experiment.

*P4, line 20: Move this sentence up to the end of the sentence (line18-20)* After rephrasing the previous sentence, we would like to keep the sentence position unchanged.

P5, line 10: change (Fig.2) to (Fig.2b)Has been changed (but to Fig. 2a → soil moisture).

P5, line 20-21: I am not quite sure if I understand this modification. Did you add a vent to the box? Then better to call it vent than inlet as the inlet is connected to the QCL. Please be more specific: I assume the 4 cm is the diameter and the 1m is the length of the vent tube and the 10 cm is the length of the foam material within the tube? What is the foam made of?

We agree with the reviewer that the box modification was not properly described. As assumed by the referee, a vent tube (to ambient air) was added, instead of a fully closed loop originally used in the Fast-Box. We described the vent tube more specifically in the revised version.

*P5, line 22: As you don't show or discuss any soil respiration measurements, I suggest to delete this information.*

EDITOR: Both referees point to this. You can avoid this criticism by including a subordinate clause mentioning the purpose. BTW avoid the tautology "CO2 soil respiration"

We kept this information in the text, because the  $CO_2$  signal (due to ecosystem respiration) was used as a proxy to check if the chamber was properly sealed (as mentioned in Sect. 2.4.3). We changed the sentence to: "The chamber was also equipped with a GMP343 CO2 probe (Vaisala, FL) to measure the soil respiration, which was used for quality control purposes (Sect. 2.4.3)."

*P5, line 24: Insert "the" after "The inflow off. . ." I assume by "the inlet" you mean " the vent"? As the FB chamber is a closed dynamic system (acc. to Hensen et al.2006).* Was changed accordingly (see also response above).

P5 line 26: please add explanations to all abbreviations used in equation 1 (V, A, Fcham)

Definitions of symbols were added accordingly.

*P6 line 12: please give the make of the thermocouple* Was added in the revised version (k type).

P7 line 24: 500; add unit Was added ("500 data points").

*P8 line 12: change (see Fig 2) to (see Fig. 2c). Please show the harvest event in the Figure (see comments to Figures)*

Was changed to "(see Fig. 2c-d)". We added the harvest event in the Fig. 2c and the fertiliser applications in Fig. 2d (see also responses to Figure comments below).

*P8 line 33: Can you give a time period for the soil temperature classes?* The time period spanned over the GOP. We included this information.

P11 line 8/9: This last sentence is not clear. Please clarify which fluxes you are talking about (individual emission source; paddock or system M/G?) We omitted this sentence as it was no more necessary in the revised version.

**Results:**

*P11 line 12: "they varied significantly"; significantly different from what? Background fluxes?* We rephrased to "...showed considerable variation...".

P11 line 13: mention the harvest, were they increased after the harvest? There was a short increase in  $N_2O$  emissions directly after the harvest event. We included this information in the text.

P12 line 15-19: In Figure 8 FU,temp and FD,temp are fluxes averaged over 3 days, while in the text you are describing average daily values (?), which is confusion. I suggest to show average daily values in Figure 8, or use different abbreviations (e.g. FU,temp3d vs FU,temp1d), or only discuss 3d averages in the text. Its not clear what the "absolute highest FUtemp" is (5117 ug N2O-N m-2h-1), if the highest average value is 660 ug N2O-Nm-2h-1.

We agree with the referee that the paragraph was a bit confusing. We rephrased it in a more concise way and only describe the 3-day averages displayed in Fig. 8.

P12 line 20: you mention that Dung related emissions "showed a relation to excreta age", please mention what kind of relation. Please change "dung patch emissions" to "dung patch fluxes" The relation can be seen in Fig. 8. We added a reference to this Figure. Additionally we changed "dung patch emissions" to "dung patch fluxes".

P12 line 25: Were the background fluxes not also sign. smaller compared to dung patch emissions? Looks like it in figure 8.

The background fluxes are also sign. smaller than dung patch emissions. We will rephrase the sentence to "...Background fluxes were on average considerably smaller than excreta fluxes".

**P12 line 32: How do you justify to set negative values to zero?**

EDITOR: Consider using a better suited function that doesn't force predicted data into the negative domain.

The formulation in the text was probably misleading. We wanted to approximate the measured 3-day averages for the dung patch fluxes (all positive, see Fig. 8) with an empirical function as simple as possible (few fitting coefficients, simple regression statistics) and with conservative assumptions concerning the extrapolation beyond the observed age period. We thus rephrased the text in the following way:

"Because the evolution of dung emissions  $F_{D,age}$  after the observed 20-day age period is unclear and a meaningful functional extrapolation was not possible, we decided to use a simple 2nd order polynomial for parameterization purposes. It allowed to reproduce the initial increase with age and a rapid decrease to zero beyond the measured age range. The fitted polynomial function is only applicable up to  $\Delta t_{EOG} \approx 25$  d, where it crosses the zero line.

P13 line 6: As "FD,temp" is not influenced by environmental conditions it equals FD (?)This should be stated here.

We include this information as suggested.

P13 line 7-14: Please define the three sectors. I suggest to insert (<0.27, 27-33, >0.33) after ". . . by three different VWC sectors." It would help to show this in a graph.

EDITOR: Consider adding a graph to an appendix

We follow the referee's suggestion and insert (<0.27, 27-33, >0.33) after "... VEC sectors". However, we do not want to add a new graph as the paper already has quite a high number of graphs and the additional information provided by a graph would be comparatively low.

P13 line 13: do you mean "similar values" or comparable to what? Can you add a stdev? Indeed, we mean "similar" values. The values including the stdev are 12  $\pm$  3 µg N2O-N m-2 h-1. We included this changes in the text.

P14, line 17: It's not clear where the grazing period ends, therefore please add this information into the table (see comments table 10b).

We assume, that the referee refers to Fig. 10b. We include the information about the grazing phases on the different paddocks in the figure.

P14 line 25-27: these two sentences are not very clear. What do you mean by variations? The magnitude of fluxes varied less? I suggest to replace "rather limited" with "less pronounced" We omitted one sentence and rephrased the text as follows:

"The up-scaled FB fluxes compared well in magnitude with the measured EC fluxes and showed a similar temporal behaviour. While generally a response to variations in environmental driving parameter could be observed, it was less pronounced for the up-scaled FB fluxes in comparison to the EC fluxes."

P15 line 2: But in Figure 11 it looks like fluxes were slightly higher for up-scaled FB fluxes. We agree with the referee on this mistake. We replaced "slightly lower" by "slightly higher".

Discussion:

Responses

**P15, line 20/21: I don't understand this sentence**

We agree with the reviewer, that the sentence was confusing and thus rephrased it as follows: "We assume that the EC fluxes are on average representative for the whole pasture system, although the contribution of the central paddocks X.11, X.12 and X.21, X.22 to the EC footprint is generally higher than the contribution of the other more distant paddocks (Fig. 4)."

P15, line 25: Include "(data not shown)" at the end of "other characteristics" as you didn't show any productivity (yield). Delete "Also" at the beginning of the next sentence. Was changed accordingly.

P15, line 28-p16 line 4: This paragraph is difficult to understand. I strongly disagree that you can conclude that an optimised diet in system M leads to a 25% reduction effect for N2O emissions based on the smaller area needed for grazing. You need to take the N2O emissions related to the maize production into account, otherwise this comparison is not valid. See answer to general comment 1 above.

P16, line 23: 1.03 kg N2O-Nha-1y-1, please explain in more detail how this value was calculated. This value should have been shown in the results section 3.3.2.

The value was calculated by using Eq. 5 (as stated) similar to the cumulative background emission in Fig. 12b (green area) but for the entire year using measured soil moisture. We specified this in the text. This full year extrapolation value was only calculated for comparison with (annual) literature values. Therefore, it does not fit into the results section, as the paper results are focused on the grazing-only period GOP.

**P16, line 26/27: This last sentence is out of context**

We rephrased the sentence to clarify the connection to the preceding text: "On pastures, background emissions may additionally result from trampling of the cows that can further stimulate the N2O production via denitrification due to soil compaction (Bhandral et al., 2007)."

**Tables:**

Table 1: As soil depth is not really a parameter I suggest to re-arrange the table; one column for each soil depth, with missing values in each column as the different parameters have not been measured in all soil layers

Since the information on the deeper layer are not really relevant for the present topic, we decided to reduce the table and only show the soil characteristics for the near-surface layer. The information on differences in the exact layer depth were moved to the footnote.

*Table 2: What does ECM stand for? Please explain abbreviation (maybe in footnote).* ECM is the 'energy corrected milk'. We added this explanation in the table caption.

Have the animals been weighted before and after the experiment? Was the weight increase considered in the calculations of the excretion (heavier animals will excrete more)?

The animals have been weighted on a daily basis and the possible weight gain is considered in the calculations of the excretions. We rephrased Sect. 2.3 and included this information. More details are given in the referenced article by VogImeier et al. (2018) for the same experiment.

Table 3: Please add information of flux measurement method (FB)Has been added accordingly.

Table 4: Please describe what the different equations are: Parameterisations of 3 day average fluxes from EB measurements, split into background, dung and urine fluxes.

A proper description of the five different equations would be quite complex and a (unnecessary) repetition of the information given in the text. This table is just used for listing the numeric values of the equation coefficients (not the equations themselves). However we added the following information relevant for the numeric values of the coefficient in the table caption: "The equation coefficients were fitted using FB chamber measurements and yield fluxes in units of  $\mu$ g N2O-N m-2 h-1. The input quantities are the soil temperature Ts (in units of °C), time since end of grazing  $\Delta$ tEOG (in units of days) and volumetric water content VWC (as dimensionless fraction)."

Table 5: I assume that you are showing cumulative fluxes. You need to mention this together with the time scale (per GOP?). I suggest to simplify the table by only having two parts; add a dotted line above the N input and to move the FB urine and FB dung fluxes above the EFs. Please add N input from dung. What does FAD stand for?

We added the information about cumulative fluxes and the time scale (GOP). Furthermore, we rearranged the table based on the referee's suggestions. We also added the N input from dung. FAD was a term used in a development version of the manuscript and was deleted.

Table 5: What is "EC integral system emission EC," ? Reading in the text (P15, line 28-31) I have the impression these fluxes are up-scaled FB fluxes to the whole system. If they are EC emissions, please describe more carefully in the text. The unit is confusing as it is an emission (concentration per area per time). Reading in the discussion I understand what you mean, but in the table it's not clear. Maybe you can explain in a foot note. "EF total" is calculated from EC, while "EF urine" and "EF dung" are calculated from up-scaled FB measurements (or not?) This needs to be stated clearly.

The mentioned value/units represented integral emissions for the entire pasture area and the investigated grazing period. In order to prevent confusion and misunderstandings, we changed the units of this type of results (also in the abstract and in the main text) and express them in units of  $N_2O$  -N cow-1 h-1, which has an equivalent meaning.

In addition we reorganized the table in a more logical way (see previous comment) and included a footnote with information on the listed quantities.

**Figures:**

*Figure 1: P29 line4 Insert "(triangles)" after "the two EC towers. . ."* Has been added accordingly.

Figure 2 b): Move the legend, or change the scale so the bars for the high precipitation events in June and July are not cut off. Has been changed accordingly.

Figure 2d): Add arrows for fertiliser application dates and for harvest date

Responses

We added the dates in Fig. 2c and 2d.

*Figure 3: Change the area showing a) to being transparent* Has been changed accordingly.

Figure 5: Please explain the reason for dotted frame. It's confusing that the lines connecting #dung and # urine patches to "Paddock flux dung patches" and "paddock flux urine patches" cross the arrows leading to "paddock flux background" and "paddock flux urine patches", it looks like they are feeding into them as well. Try to show clearer (maybe with a curved line over the crossing line). The dotted frame indicated a further processing step. We realized that this is actually not needed and changed it to a standard frame. We agree with the reviewer, that the line crossings are a bit confusing and thus updated the graph (less crossings, use of curved line).

Figure 6: Add arrows for exact fertilisation and harvest events. Add the date of the skipped value. It would be good to include the information of grazing periods in the graph to explain the increased fluxes. We added vertical lines in the graph indicating fertilization and harvest events, and we added the date of the skipped value in the caption. Furthermore, we added the information that for the analysis of grazing related emission, the non-shaded periods of Fig.6 were used. Detailed information of the rotational grazing regime is already displayed in Fig. 3c.

Figure 7: I suggest to change the unit to ug N2O-N m-2h-1, it makes it easier to read the values in the graph and to compare to values you describe in the text (for Figure8, where ug N2O-N m-1h-1 are used). In the legend insert "from different sources" after "the comparison of fluxes.." and add the information that the fluxes were measured with FB technique.

We agree with the referee and changed the units accordingly. We also added the information on the flux measurement technique in the caption.

Figure 8: Same comments about units as for Fig. 7. Please add in legend that fluxes were measured with FB. Are there any standard errors for the background fluxes, or were they too small to be seen? I suggest to change to x-axis description to "age of excreta [d]". Give information about the fitted curves (refer to equation 3+4).

We changed the units and the x-axis description to "age of excreta [d]" and we added a reference to Eq. 3, 4.

There are standard errors for the background fluxes, but these are smaller than the symbols for the averages. We added this information in the legend.

*Figure 9: Same comments about units and mentioning that FB method was used as for Fig. 7.* We changed the units as requested.

Figure 10: It would help to add the grazing period in either Fig 10 b) or c) We added the grazing periods in Fig. 10b.

**Referee #2**

**General comments**

The authors used gap filling approaches to fill gaps in their eddy covariance N2O flux dataset but there was not much discussion about the gap filling results. My suggestion is that this discussion should be expanded.

We used only one approach (LUT) for the gap filling of our EC fluxes, as mentioned in Section 2.5.3. This approach was chosen based on the cited evaluation by Mishurov and Kiely (2011) who discussed different gap filling approaches in more detail. It was not the objective of this study to assess the performance of gap filling approaches. However, we used the variability between LUT and three other approaches for estimating the uncertainty of the gap filling procedure.

In addition, it would be important to include more details in the methodology on the EC and chamber measurements and scaling approaches (see specific comments). Some sentences in the text are difficult to understand, so the writing requires further work. I also noticed some grammar mistakes in the text. I recommend the authors to perform a thorough review of the manuscript to correct these mistakes before resubmitting the manuscript.

We think, the methodology on the EC and chamber measurements (including scaling approaches) is already described very extensively with about 6.5 pages (excluding figures). Nevertheless, we included most of the specific comments on methodology issues (see answers to specific comments below). Regarding the language related issues, we performed a thorough review of the text and corrected a larger number of language mistakes.

**Specific comments**

L12 – replacing "season 2016" by "season of 2016". In addition, I suggest including the number of dairy cows for each herd.

Was changed accordingly. We also included the number of dairy cows (12 for each herd).

L15 – "Excreta patches and background surfaces on the pasture were identified manually". I suggest to be more specific here by saying that urine patches were identified based on the soil electric conductivity.

We rephrased the sentence to "After different grazing rotations, background and urine patches were identified based on soil electric conductivity measurements while fresh dung patches were identified visually."

 $L20 - "(960 \pm 219 \text{ g N2O-N}, \text{ or } 25 \%)"$  This number is a little confusing. What does the 25% represent and shouldn't the emission units be expressed in per area?

We agree with the reviewer that the number was confusing. Therefore we removed this quantitative statement from the abstract. In order to prevent confusion and misunderstandings, we changed the units of this type of results in the main text and expressed them in units of N2O -N cow-1 h-1, which has an equivalent meaning. Correspondingly we revised the phrasing related to the comparison of the two systems.

L29 – replace "In the atmosphere, nitrous oxide" by "Nitrous oxide". In addition, include the appropriate citation for this sentence.Was changed accordingly. Additionally, we added a reference to the IPCC (2014) report.

L30 – replace "it has a strong potential" by "N2O has a strong potential". I noticed that the replacement of nous by pronouns in some sentences throughout the text can compromise the clarity

of those sentences. I suggest the authors to be as direct as they can in their sentences for the sake of clarity.

Was changed accordingly. Furthermore, we tried to locate those replacements of nouns and changed them to the proper nouns.

**Page 2**

L1 – "especially by cows". Are you referring specifically here to dairy cows? If so, please specify. No, we refer to cows in general.

L3 to L5 – "Directly applied on a pasture soil. . ." this sentence is awkward and needs to be reworded. We reworded the sentence to:

"The available reactive N is used by microbial nitrification and denitrification processes where significant amounts of N2O can be produced."

L17 – replace "(e.g. EF of 0-14% of applied urine N, n=40; Selbie et al., 2015) and many of those studies measured the" by "(e.g. EF of 0-14% of applied urine N, n=40; Selbie et al., 2015). Many of those studies measured the". In addition, give some examples of the "many of those studies". We rephrased the sentence to:

"However, the range and thus the uncertainty of specific urine EFs is rather large (0-14%, n=40) as shown by Selbie et al. (2015) based on a survey of literature reports. Many of those studies measured...".

L20 – "these emissions" which emissions?

We rephrased the sentence to make it clear that we meant the emissions associated to animal excreta. "The efficient use of fed N is essential to reduce the emissions associated to animal excreta."

L20 - "(e.g. Arriaga et al., 2010)" provide more examples of studies and more the citation to the end of the sentence.

We provided more examples of studies and we put the citations to the end of the sentence. "...N excreted by the animals (e.g. Yan et al., 2006; Arriaga et al., 2010; Dijkstra et al., 2013)."

L24 – "real practice conditions". Do you mean real management conditions? We rephrased the sentence as follows: "...but corresponding emission experiments under real grazing conditions for a full season, to our knowledge, have not been reported hitherto."

L23-24 – "experiments. . . are very rare". Cite some of the existing ones. Actually, to our knowledge, no comparable experiment exists. We rephrased the sentence as shown in the previous comment.

L26 – "and to attribute them to certain emission drivers" this statement needs to be reworded for clarity.

We rephrased the sentence to "...to attribute the measured fluxes to potential emission drivers...".

L30 – "by integration over a larger domain". Integration of what? Do you mean fluxes? Larger domain than chambers?

We rephrased the sentence to "...integrating over multiple emission sources over a larger spatial domain."

**Page 3**

L8 to 9 - "We aimed at a better understanding of the quantity of the overall pasture emissions, the different emission sources and the reduction of corresponding uncertainties". This sentence is awkward and needs to be reworded.

We omitted this sentence, as it is redundant.

L12 to 13 – provide the experimental period. We provided this information (grazing period 2016).

L14 – "annual average rain amount". Is snow also included in the total amount? If so, replace the word "rain" by "precipitation". Snow is also included, thus we changed "rain amount" to "precipitation".

L15 to 16 – "(about 20 % clay, 35 % silt and 45 % sand" there is no need to show this since this soil texture data are shown in Table 1. Was changed accordingly.

L16 – "Soil measurements were performed. . .". Can you be more specific? This sentence is referring to the preceding sentence (with reference to Table 1). We rephrased the sentence to: "Soil profile samples for analysis of texture and other soil characteristics were taken at four locations on the pasture in 2013 and 2016."

**L19 to 20 – "the fertilization rate was in the order of 120 kg N ha-1 per year between 2007 and 2015". Can you please specify the fertilization timing?**

EDITOR: Please reconsider you answer. Are the background emissions during the campaign not depending on the timing of previous fertilizations? Sometimes a small amendment of information can make the article relevant for other work, e.g. reviews, which will increase its scientific value. We indeed have discussed in Section 4.2, that the background emissions may (partly) be attributed to fertiliser application in previous years. Therefore we added the information about the average annual fertiliser application rate of the previous years. However we cannot see the benefit of just adding fertiliser timings for the previous years (e.g. for modellers or synthesis/review studies) without additional detailed information about the management and the weather conditions. But that would be out of proportion to the scope of this study. We are planning to publish a multiple-year comprehensive dataset including metadata of the study site on a flux data repository.

L23 – "12 cows per system.". Please reference figure 1. We included a reference to Fig. 1a.

L24 – "with additional maize silage". Was this silage offered to the cows in a different area? Did the silage supplementation influenced the time in which the cows spend in the grazing system? The silage was fed in the barn when the cows had to go there for milking twice a day (this information was added in the revised manuscript). In order to avoid an influence of the supplement feeding on the grazing time, the barn and grazing times were always fully synchronous for both herds/systems.

L30 – "X indicating both systems". I suggest using M or G instead of X to avoid confusion. We would like to keep the "X" because it simplifies the text considerably when referring to both systems equally.

**Page 4**

L15 – "For the comparison with the field-scale EC". Which comparison? Be more specific. We agree with the reviewer that the sentence is not specific enough. We rephrased the sentence to "The comparison between the field-scale EC method and the small scale chamber measurements also required estimates of the number of dung and urine patches on the pasture."

L1 to 2– "Conductivity values exceeding a threshold of 0.15 mS cm-1 were marked as possible urine patches for further chamber measurements." It is important to explain how this electric conductivity threshold was established.

We added more information in the text. The threshold of 0.15 mS cm-1 was chosen based on preexperimental tests with artificially applied urine patches on the pasture and areas not affected by grazing for a few month (background). The value of 0.15 mS cm-1 was determined as the maximum of the observed background conductivity, and it was still far below the observed conductivity of fresh urine patches (see also Fig. 3).

**L10 – "taken mainly during dry soil conditions" Can you provide the soil water content associated with "dry soil conditions"?**

There is no fixed water content threshold for dry soil condition. Nevertheless, as can be seen in Fig. 2 and Fig. 9, we refer to volumetric soil moisture contents below roughly 0.4, thus clearly lower compared to the ones before July. We added this information in the revised manuscript.

L18 – "a 40 m 1/4" PA tube allowing". Use metric units do express the dimensions of the tubing. Does 1/4" refer to the internal diameter of the tube? Please specify. What does "PA" stand for? We added some information as follows: "The sample air was drawn continuously from the FB headspace through a 40 m 1/4" polyamide (PA) tube to the analyser …". We would like to keep the tube diameter in inches, as this is the official commercial labelling of this product.

L19 – "The sample flow rate Q was typically around 8 l min-1". Did you use a mass flow controller to keep the flow rate constant?

No, we did not use a flow controller. The flow rate was controlled by the controlled pressure (30 Torr) in the QCL analyser cell and a flow restrictor needle valve at the QCL inlet. The inlet tube represented an additional flow resistance. Since the effect of the valve and the tube were constant over time, the flow also remained quite stable.

L21 – "foam material to avoid uncontrolled air exchange". Was the chamber covered with some insulating material? What was the typical temperature differences within and outside the chamber during these measurements?

No, the chamber was not covered with insulating material. Nevertheless, as the measurements with a fast-box are very quick (typically within 2 minutes), the temperature differences between inside and outside the chamber stayed typically below 0.5 °C. When starting a new measurement, the chamber volume was always flushed (by opening/tilting the box by 90° until the chamber volume was completely mixed with the ambient air).

L21 to 22 – "The chamber was also equipped with a GMP343 (Vaisala, FL) CO2 probe to measure the soil respiration." Do you show this CO2 data? If not, I suggest excluding this sentence. EDITOR: Both referees point to this. You can avoid this criticism by including a subordinate clause mentioning the purpose. BTW avoid the tautology "CO2 soil respiration"

We kept this information, as the  $CO_2$  concentration increase (related to the soil respiration) was used as a proxy to check if the chamber was properly sealed (as mentioned in Sect. 2.4.3). We added this information in the text.

L22 to 23 – "The increase in concentration after placing the chamber on the soil was recorded every three seconds for a time period of about 90 seconds." For your chamber flux calculations, did you take into account the time necessary to purge this long tube right after the sampling line was connected to the analyzer?

Yes, this time was of course taken into account. We added this information in the sentence.

L2 – "(slow chamber volume exchange and short measurement time)". Can you provide an average value for the chamber volume exchange?

The average volume exchange time is a direct function of the given chamber dimensions and flow rate. It was about 40 min. We added this information in the text.

L13 to 14 – "a thermocouple for air temperature measurement within the chamber, a GS3 probe (see Sect. 2.4.1) and a ML3 Thetaprobe (Delta-T Devices Ltd, UK) for soil moisture and temperature observations (c. 0-5 cm and 0-10cm depth, respectively)." This sentence is a little confusing and needs to be reworded.

We reworded the sentence as follows: "a thermocouple (type K) for air temperature measurement within the chamber, a GS3 probe (see Sect. 2.4.1) for soil moisture, soil temperature and soil conductivity measurements (c. 0-5 cm depth) and a ML3 Thetaprobe (Delta-T Devices Ltd, UK) for soil moisture and soil temperature observations (c. 0-10 cm depth).".

**Page 7**

*L4* – "were fenced to avoid unwanted animal contact". Can you provide the area of the fenced area around the tower?

The fence was in a distance of about 2 m around the tower in the main wind direction sectors. Only in the direction where the analysers were located in an air-conditioned trailer/container, a larger area was fenced (see white area around EC tower positions in Fig. 1a). We added the information with the fenced radius around the EC tower.

L9 – Does this sonic anemometer infers the air temperature based on the sonic temperature or it has its own temperature sensor?

It should be quite clear that a 'sonic anemometer-thermometer' infers the wind vector and the temperature from speed-of-sound (sonic) measurements.

EDITOR: Does this mean that you have recalculated the air temperature from the sonic temperature? This is prone to errors, because some sonic anemometers have obvious problems to give accurate temperature readings. Did you check your sonic anemometer for such errors?

We agree with the editor that such temperature measurements can have accuracy problems. We compared the sonic temperature to the weather station temperature and roughly checked the quality of the sensible heat flux by checking the energy budget closure and found no obvious problems. In the present study, we use the sonic temperature measurement only very marginally (and indirectly: e.g. in the spectral correction) and therefore a detailed assessment of its quality in the manuscript would not be adequate.

**L11 – Please provide the pore size of the filters**

The Midisart 2000 has a filter pore size of 0.2  $\mu m$  and the AcroPak has a filter pore size of 0.2  $\mu m.$  We included this information in the text.

**L16- "The sample frequency of the EC system was generally 10 Hz". Does this mean that there was variation in the sample frequency? Why is that?**

We agree with the reviewer about the confusing phrasing. The EC system was always operated at 10 Hz. We omitted "generally".

**L18-19 – This sentence is awkward and needs to be reworded.**

We reworded the sentence to "Additionally the program visualized the measurements of the  $N_2O$  concentrations and fluxes, calculated with a preliminary online flux calculation. The program also allowed to check the EC system by remote access."

L22 – "The approach is based on. . ." What approach are you referring to?

We refer to the customized program mentioned in the previous sentence. We rephrased the sentence accordingly.

**L24 – 500 data points?**

Yes, we meant 500 data points (added in the text).

**L28 – "several seconds". Provide the typical time lag value and its standard deviation.**

The typical time lag was about 6 seconds for system M and about 7 seconds for system G. We added this information in the manuscript. But it is not possible to give a meaningful standard deviation of the 'dynamic' lags determined by the peak position in the cross-covariance function. In many cases, the signal-to-noise ratio of the fluxes was small due to low emissions or non-stationarity. In these cases the 'dynamic' lags were often not meaningful and very large. Due to this reason we applied a lag window filter (as described in Sect. 2.5.2) and used an average default lag otherwise. Thus the selected good quality lags were by definition within a window of ±0.61 s.

**L31 – "a time window of 0.61 seconds". How was the number determined**

The number was determined empirically based on the variability of the determined dynamic lag times.

**Page 8**

L1 – "In order to minimize the effect of non-stationarities in the time series, the 30 min flux was finally calculated as average over six 5 min subinterval flux values.". I wonder what would be the effect of this averaging approach on the low frequency spectral losses of their EC system. Furthermore, if you are already screening the data for non-stationarity (page 8 L24) why to estimate fluxes for these short time intervals?

The low frequency losses were in the range of 1-5 %, based on theoretical calculations (Kaimal cospectra and transfer function for block averaging) as well as on the comparison of 30 min fluxes and 5 min subinterval fluxes. The theoretical approach was used to correct the fluxes for this low-frequency damping effect. This information was added in the manuscript.

We now recognized that we erroneously mentioned the application of a stationarity criteria in the EC method description. We actually did not apply a stationarity filter for the fluxes. As mentioned by the referee, this would not make much sense (and it did not have a significant effect) in combination with the minimizing of non-stationarity effects by using average 5-min subinterval fluxes. We removed the respective statement from the manuscript.

EDITOR: In your answer you mention that the use of 5 minutes measurement intervals lowered the effects of non-stationarity. Did you test this? And if yes, which of the 5 minutes intervals did you choose to represent the 30 minute average? If you used them all and averaged them, can you explain why that still reduces the effect from non-stationarity at the 30 minute time scale? Wouldn't you rather need to test whether the 5 minute interval was stationary and then give high-pass corrected flux values from all 5 minute covariances? A small self-critical discussion of this procedure is advised. As stated in the text, we averaged all six 5-min subinterval fluxes to 30 min (without discarding any subinterval). The reduction of non-stationarity effects by this procedure is a fundamental assumption in the common flux stationarity test (Foken et al., 2012). In the spectral space, the use of shorter averaging intervals cuts off the lowest frequencies of the normal 30-min flux spectra, which largely contain the non-stationarity effects (red noise). As mentioned above, this also leads to a very moderate damping of the turbulent cospectrum, which was corrected for. It should be noted that the 5-min flux averaging time, as used here for a measurement height of 2 m, has about the same low-frequency damping effect as a normal 30-min flux averaging for a measurement height of 12 m (e.g. above a forest).

We indeed observed a significant reduction of the non-systematic (red noise type) variations in the aggregated 30 min averaged fluxes by this procedure, so that an additional stationarity filtering had only little effect and was therefore not used. We added this finding in Sect. 2.5.3.

L7 – "half-hourly damping factors". Do you mean dampening factor?
L9 – "damping factors" see comment above
L10 – "damping effect" see previous comment
EDITOR: It looks as if damping was the more accurate term.
We think, both terms can be used, but the term "damping" is more commonly used in the EC
literature. Therefore we kept it.

L19 – replace "which often result" by ", which often resulted" We prefer to use "which often result" to indicate, that this not only happened in the past but is an ongoing issue with EC measurements.

**L28 – "It was driven". What is "it" referring to?**

The sentence was rephrased to "The occurrence of data gaps showed a diurnal pattern with stronger data loss during the night, which was driven by the wind pattern with typically stronger wind speeds during daytime and calm nights."

**Page 9**

L15 – "and it has to be checked". What is "it" referring to?We are referring to the spatial dimension of the footprint. This has been changed in the revised version.

L22 – "80' 000 trajectories were released backwards in time" replaced by "80,000 fluid particles were released backwards in time". Also, what is the time scale of this simulations? 30-min periods? Yes, the footprint simulation time scale was 30 min. We added this information in the text and replaced trajectories by fluid particles.

L24 – "systematic uncertainty". Do you mean "accuracy"?We mean "systematic uncertainty" as described in the referenced articles at the end of the sentence.

L1 – I think section 2.6 is out of place. It should come after section 2.7. We think, it is actually not out of place as Sect. 2.7 builds on data retrieved from Sect. 2.6 (soil moisture / soil temperature measurements). Thus, we kept the structure of the sections.

L2 – what is the datalogger model used in this study?

The automated weather station was equipped with a Campbell Scientific CR10X data logger. We added this information in the manuscript.

L6 – In this section, it would be important to provide the spatial resolution of the grid used for upscaling the chamber fluxes. More details are also necessary on how the authors went from the output of Eq. 2 to the scaled fluxes. Did you generate digital maps of source emissions and then overlapped these maps with a footprint map? What was the software used to do these calculations? EDITOR: You forgot to mention the software.

We are not completely sure, whether we understand the question of the referee. We did not use a grid to upscale the chamber fluxes to the EC system. Equation 2 was evaluated for each paddock (integrating the particle touchdowns within the respective paddock area) for each 30 min interval. This resulted in a footprint contribution for each paddock which was multiplied by the paddock scale emissions for urine, dung and background as described in Fig. 5 and Section 2.7.

We improved the description of the upscaling procedure in Sections 2.5.4 and 2.7 in this respect. We used the statistical software R (R Core Team, 2016) for these calculations (and generally for all major calculations). We added this information in the text.

**Page 11**

L10 – "Occasional negative individual flux values". What is the detection limit of this EC system? I think this would be an important variable to know to interpret these fluxes.

The negative fluxes exclusively resulted in cases, when no peak in the cross-covariance function could be identified (and thus the value at the default lag was used). Thus is can be concluded that the negative fluxes were generally not statistically significant, i.e. below the detection limit (which was time dependent e.g. due to the varying influence of non-stationarity effects). We rephrased the text and added this information in the manuscript.

**Page 12**

L3 - "Fluxes of background and dung patches were significantly smaller". Did you perform a statistical test to support this statement?

Yes, we performed the Student's t-test which resulted in p-values < 1e-12. But we also think that the results plotted in Fig. 7 are very clear in this respect. We rephrased the sentence to indicate, that we actually meant the absolute value of the fluxes (and not only smaller in a statistically calculated way).

**Page 14**

L25 – "the variations were less pronounced". Which variations were less pronounces. We rephrased the sentence to "...the variability of the up-scaled FB fluxes were less pronounced.".

**Page 15**

L 19 – "The good agreement between the two independent approaches" provide a statistical index to support this statement.

We expanded the sentence to: "The good agreement with a relative difference below 1.5 % for yearly sums (which is far below the uncertainty range, see Table 5) between the two independent approaches...".

L28 to 29 – This sentence is a little confusing and needs to be reworded.

We rephrased the sentence as follows: "For assessing the effect of the N reduced diet on excreta related N2O emissions, the emissions per cow and grazing hour have to be compared".

L21 – "significant system difference". Over which period of time and shouldn't this difference be expressed per area?

We agree with the reviewer and replaced this value by the emissions per cow and grazing hour of system M and G (see comment to Page 1, line 20 above).

**Page 16**

L3 to 4 - "e.g. N2O emissions related to the maize production. . .". Could you include values in the literature typical emission factors for corn silage production? These data would allow a fair comparison between the two grazing systems.

EDITOR: I agree that one can easily think of the evaluation of the upstream processes or the full production chain. Therefore, it must be made very clear what the (limited) scope of this manuscript is, but in the discussion, it the restricted scope should be mentioned again and briefly discussed what else would be needed to evaluate the whole chain. This is necessary for the readers not to draw wrong conclusions. The conclusions should also refer very clearly to the limited scope of the study to avoid any misinterpretation.

It needs to be noted here, that we only considered the  $N_2O$  emissions related to the cow excreta on pasture in this study (as indicated by the title). We revised and clarified the text in this respect in various parts of the manuscript (see also response to General Comment 1 of Referee#1).

We rephrased the sentence to make clear that a full accounting of the production chain on N2O emissions would require much more complex calculations (not just the inclusion of the maize production:

"Any further N2O emissions e.g. related to fertiliser application on the pastures or for the supplement maize production were not taken into account here. A comparison of entire production systems would require many additional assumptions outside the specific scope of this study. It also has to be considered that the N optimisation of the diet is not necessarily linked to the supplemental feed of arable crops like maize, but may as well be achieved with different feed strategies (e.g. grass varieties with a high content of water soluble carbohydrates; Misselbrook et al., 2013)."

**L9 – "They are based on". Specify who are "they".**

We meant the EFs. We rephrased that in the revised version of the manuscript.

**Page 17**

L23 – "emission optimum". What does the word "optimum" mean here? Low N2O emissions? EDITOR: Your answer does not allow to judge whether you will adopt writing what you mean or what Butterbach Bahl et al. (2013) wrote, i.e. contrary to what you mean. Please be accurate in your responses.

No, we meant emission maximum. The term "optimum" is often used in this context (e.g. Butterbach-Bahl et al., 2013). We changed the term "optimum" to "maximum".

**Page 28**

Table 5 – "EC integral system emission EC". Do you mean: Integral EC flux system emission? The entry represents integral emissions for the entire pasture area and the investigated grazing period. In order to prevent confusion and misunderstandings, we changed the units of this type of results (also in the abstract and in the main text) and expressed them in units of N2O-N cow-1 h-1, which has an equivalent meaning.

**1 Grazing related nitrous oxide emissions: from patch scale to field scale**

2 Karl Voglmeier1,2, Johan Six2, Markus Jocher1, Christof Ammann1

3 1Climate and Agriculture Group, Agroscope, Zürich, 8046, Switzerland

[revised manuscript text omitted]